# Novel functions for integrin-associated proteins revealed by analysis of myofibril attachment in *Drosophila*

Hannah J Green[1,2,3], Annabel GM Griffiths[1], Jari Ylänne[2,3], Nicholas H Brown[1]*

[1]Department of Physiology, Development and Neuroscience, University of Cambridge, Cambridge, United Kingdom; [2]Department of Biological and Environmental Sciences, University of Jyväskylä, Jyväskylä, Finland; [3]Nanoscience Center, University of Jyväskylä, Jyväskylä, Finland

**Abstract** We use the myotendinous junction of *Drosophila* flight muscles to explore why many integrin associated proteins (IAPs) are needed and how their function is coordinated. These muscles revealed new functions for IAPs not required for viability: Focal Adhesion Kinase (FAK), RSU1, tensin and vinculin. Genetic interactions demonstrated a balance between positive and negative activities, with vinculin and tensin positively regulating adhesion, while FAK inhibits elevation of integrin activity by tensin, and RSU1 keeps PINCH activity in check. The molecular composition of myofibril termini resolves into 4 distinct layers, one of which is built by a mechanotransduction cascade: vinculin facilitates mechanical opening of filamin, which works with the Arp2/3 activator WASH to build an actin-rich layer positioned between integrins and the first sarcomere. Thus, integration of IAP activity is needed to build the complex architecture of the myotendinous junction, linking the membrane anchor to the sarcomere.
DOI: https://doi.org/10.7554/eLife.35783.001

*For correspondence:
nb117@cam.ac.uk

Competing interests: The authors declare that no competing interests exist.

## Introduction

Cell adhesion to the extracellular matrix (ECM) is essential for the development and homeostasis of multiple cell types and tissues (*Winograd-Katz et al., 2014*). Adhesion to the ECM is primarily mediated by integrins, α/β heterodimeric transmembrane receptors. The medical importance of integrin adhesion is exemplified by the range of human diseases resulting from its loss or misregulation (*Winograd-Katz et al., 2014*). The extracellular domains of both integrin subunits binds ECM ligands, whereas it is primarily the intracellular domain of the β subunit that recruits intracellular integrin-associated proteins (IAPs). As β subunit cytoplasmic tails lack enzymatic activity and almost all are short (~47 residues) it is the IAPs that mediate integrin signalling and interaction with the actin cytoskeleton (*Campbell and Humphries, 2011*). Over 150 different proteins have been implicated in integrin adhesion, and 60 are consistently found in adhesions (*Horton et al., 2015*). We aim to discover why so many proteins are required for the seemingly simple task of linking the ECM to the actin cytoskeleton. Two features of integrin adhesion sites may help explain the complexity of the machinery required: their complex architecture, and their ability to respond to mechanical forces.

Super-resolution microscopy of IAPs revealed that focal adhesions in fibroblasts have a complex nano-architecture with distinct layers in the Z-axis: a membrane proximal integrin signalling layer, a force transduction layer and an actin regulatory layer that connects to the actin stress fiber (*Kanchanawong et al., 2010*). Most IAPs localise to specific layers, but talin spans two layers with its N-terminus in the integrin signalling layer and C-terminus in the force transduction layer. It is not yet known whether a similar nano-architecture is present in stable integrin adhesions within tissues. In *Drosophila* embryonic muscles talin is orientated similarly to fibroblasts, whereas in the developing

**eLife digest** Our body consists of many different types of cells that build our tissues and organs. To do so, cells need to be able to stick together. One family of proteins called integrins helps to keep cells connected. They sit across cell membranes and anchor cells to the networks of protein fibres outside cells that link and strengthen our organs, and also connect muscles to tendons.

In fruit flies, the indirect flight muscle attach to the thorax of the insect, and create wing movements by 'deforming' the thorax. These flight muscles resemble the muscles of animals with a backbone, and consist of many different fibres. At the end of these fibres is a plaque of a protein important for muscle contraction, known as actin. Integrins attach to these actin plaques, allowing the ends of the muscle to anchor to the tendon.

Integrins form complexes with so-called 'integrin-associated proteins' inside the cell, which regulate integrin. Integrins and integrin-associated proteins are essential for proper muscle development, but until now it was not fully understood how they interact with each other. Here, Green et al. explored the role of some of these proteins in the indirect flight muscles of fruit flies.

This revealed that the connection between muscle and tendon is a balancing act. Some integrin associated proteins boost the attachment, whilst others block it. One protein, tensin, increased integrin attachment, whilst another, FAK, blocked tensin, decreasing attachment. Similarly, a protein called PINCH expands the attachment, whilst a protein called RSU1 reduced the activity of PINCH to the correct level. Moreover, the end point of the muscle fibres was discovered to be composed of four distinct layers, including a newly identified layer of actin, which is built by three other integrin-associated proteins.

The flight muscles of fruit flies are similar in structure to the skeletal muscles that move our own limbs. An important next step is to discover whether these integrin-associated proteins work similarly in our muscles. A better understanding of how they work together could help with research into diseases of the muscles.

DOI: https://doi.org/10.7554/eLife.35783.002

wing both ends are close to the membrane (*Klapholz et al., 2015*). This suggests diverse architectures of integrin adhesion sites, and an additional rationale for the large IAP number.

The maturation and stability of focal adhesions in vertebrate cells in culture requires force from actin polymerisation and/or acto-myosin contraction (*Sun et al., 2016*). Building integrin-mediated attachment sites in muscles during embryogenesis has two phases: the initial attachment phase that does not require myosin-mediated contractions, followed by an increase in IAP levels and stoichiometry changes, as muscles begin contracting (*Bulgakova et al., 2017*). Such a force-responsive mechanism may ensure adhesion strength is balanced with contractile force. The recruitment of vinculin by talin is a valuable paradigm for how adhesion can be strengthened in response to force, with force-induced stretching of talin unfolding protein domains to reveal binding sites for vinculin, and bound vinculin providing new links to actin (*Klapholz and Brown, 2017*).

A valuable way to assess the contribution of each IAP is to compare the defects that occur when they are genetically removed. The reduced number of paralogs has made this comparison easier in *Drosophila*. Focusing on integrin function at muscle attachment sites, analogous to myotendinous junctions, all IAPs are present but the consequence of removing IAPs ranges from complete loss of integrin adhesion to no apparent defect (*Bulgakova et al., 2012*). The loss of some IAPs could be tolerated either because they contribute a minor function not necessary in the laboratory, or due to redundancy between IAPs. To discover how these apparently minor IAPs contribute to integrin adhesion, and test their redundancy, we utilized an integrin adhesion that resists the strenuous activity of flight.

Indirect flight muscles (IFMs) are the major muscles in the adult thorax that power flight (*Gunage et al., 2017*). IFMs are more similar to vertebrate muscles than most *Drosophila* muscles, because IFMs are fibrillar as opposed to tubular muscles such as the adult leg muscle. Fibrillar muscles are made up of myofibers, which in turn contain hundreds of myofibrils. The myofibril sarcomeres are of standard structure, comprising a regular repeating structure of overlapping actin and

myosin filaments. Actin filament barbed ends are anchored at Z-lines with their pointed ends towards the center of the sarcomere, while bipolar myosin filaments overlap with actin filaments in the center of the sarcomere and are anchored at M-lines. In the IFMs, sarcomeres form simultaneously along the length of each myofibril during pupal development (*Weitkunat et al., 2014*) and then elongate by growth from the actin pointed ends, and widen by circumferential addition (*Mardahl-Dumesnil and Fowler, 2001*; *Shwartz et al., 2016*). At the ends of each myofibril are modified terminal Z-lines (MTZ), which contain sites of integrin adhesion to the epidermal tendon cell. The myofibril and tendon cell membranes form finger-like protrusions that interdigitate with one another (hereafter referred to as interdigitations). The attachment sites in IFMs appear by electron microscopy more complex and larger than those in larval muscles, with ~1 μm separating plasma membrane from first sarcomere versus 0.1–0.5 μm (*Reedy and Beall, 1993*; *Tepass and Hartenstein, 1994*). Furthermore, IFMs contract 200 times per second (*Fry et al., 2003*), whereas larval muscles contract once every ~5 s during crawling (*Crisp et al., 2008*).

We use the IFMs to discover functions for four IAPs (tensin, RSU1, focal adhesion kinase (FAK) and vinculin) that surprisingly are not needed for viability of the fly, despite all colocalizing at integrin adhesion sites throughout development, and being conserved in all metazoa (*Alatortsev et al., 1997*; *Grabbe et al., 2004*; *Kadrmas et al., 2004*; *Torgler et al., 2004*; *Maartens et al., 2016*) (diagrammed in *Figure 1—figure supplement 1A*). We recently reported a new phenotype in flies lacking vinculin (*Maartens et al., 2016*), consisting of defective actin organisation in IFMs, suggesting that IFMs might reveal functions for other IAPs.

Here we show that indeed IFMs reveals a new phenotype for each IAP examined, and genetic interactions reveal how they work together by balancing positive and negative control. Tensin is able to bind integrins (*Haynie, 2014*), and contributes to focal adhesion maturation (*Rainero et al., 2015*) and integrin inside-out activation (*Georgiadou et al., 2017*). In IFMs we also find tensin elevates integrin activity, and discover this is negatively regulated by FAK, a cytoplasmic tyrosine kinase (*Kleinschmidt and Schlaepfer, 2017*), thus providing a new mechanism for its inhibition of integrin adhesions (*Ilić et al., 1995*). Like FAK, we discovered that RSU1 has a negative regulatory role in IFMs. RSU1 is a component of the IPP complex, containing integrin-linked-kinase, PINCH and parvin, via its binding to PINCH, and the function of this complex is not well understood (*Xu et al., 2016*). Here we show that instead of promoting IPP function, as previously thought (*Kadrmas et al., 2004*; *Dougherty et al., 2005*), RSU1 inhibits PINCH activity. By examining the localization of different IAPs and actin-binding proteins, we show that the MTZ is composed of 4 distinct layers. One of these layers, an actin rich region between the actin layer adjacent to the membrane and the start of the first sarcomere, requires vinculin, the mechanosensitive actin-binding protein filamin (*Razinia et al., 2012*) and the nucleation promoting factor WASH (*Alekhina et al., 2017*). Finally, we find that the mechanosensitive protein vinculin has dual functions at IFM adhesions. As mentioned, vinculin recruitment by stretched talin is a paradigm for mechanotransduction, yet we find that vinculin, once opened, can also function independently of talin, building an actin zone by promoting mechanical opening of filamin.

## Results

### Indirect flight muscles (IFMs) reveal phenotypes for viable IAP mutants

To understand how IAPs contribute to integrin-mediated adhesion we sought to identify defects that occur in their absence. An unexpected finding is that some highly conserved IAPs can be genetically removed without impairing viability or fertility. Thus, we have been searching for biological processes that fail in the absence of this group of IAPs. We recently discovered that whereas the genetic removal of vinculin does not impair viability or fertility (*Alatortsev et al., 1997*), it does cause a fully penetrant phenotype in the highly active IFMs in the adult (*Maartens et al., 2016*). This prompted us to examine the IFMs in flies lacking three additional IAPs that are also present at integrin adhesion sites throughout development yet can be removed without impairing larval muscle function or viability: tensin, RSU1 and FAK (*Grabbe et al., 2004*; *Kadrmas et al., 2004*; *Torgler et al., 2004*) (although not required for viability, loss of RSU1 and tensin causes wing blisters).

We analysed the development of the myofibrils in each muscle by staining for filamentous actin at different stages of development and imaging by confocal microscopy (*Figure 1A*). No gross defects

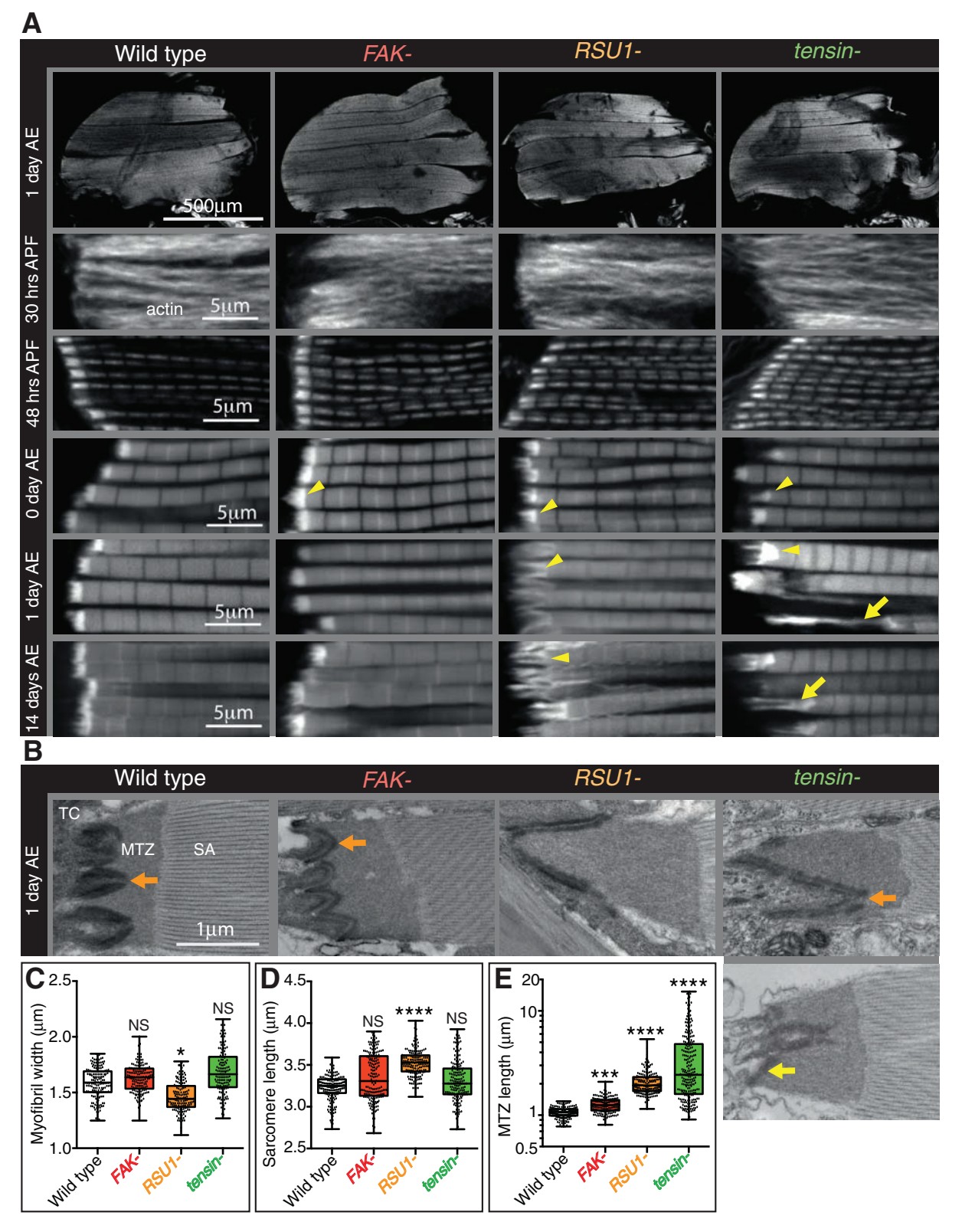

**Figure 1.** Indirect flight muscle (IFM) attachment sites reveal phenotypes for viable IAP mutants. (A–E) Examination of IFMs in the absence of FAK, RSU1 and tensin. Samples were fixed at the stage indicated, stained for F-actin (white, using phalloidin in this and all subsequent figures), and imaged by confocal microscopy (in all cases except *Figure 5*). (A) Defects in MASs arise at eclosion of the adult from the pupal case (0 hr after eclosion (AE)),
*Figure 1 continued on next page*

Figure 1 continued

but are not observed during pupal stages (30 or 48 hr after pupae formation (APF)). Yellow arrowheads indicate myofibrils with elongated MTZs and yellow arrows indicate detaching myofibrils. Quantification of MTZ length is shown in E. (**B**) TEM images of MTZs at the indicated stage and genotype, showing MTZ, tendon cell (TC) and sarcomeric actin (SA). An electron dense region is observed close the plasma membrane, which is absent at the base of interdigitations in wildtype MTZs (11/12) (orange arrow) but is continuous in MTZs lacking FAK (9/10). In attached myofibrils, two electron dense lines are seen at the plasma membrane, whereas in detached myofibrils only one line is seen (yellow arrow). Note the disappearance of fine interdigitation in RSU1- and tensin -. (**C–E**) Quantitation of muscle parameters in adults 1 day AE: in these and all subsequent figures, for each genotype 10–30 myofibrils were measured from each of 10 individuals; all data points are shown, overlain with a boxplot showing median, interquartile range, minimum and maximum; statistical test used was Mann-Whitney U. *p≤0.05; **p≤0.01; ***p≤0.001; ****p≤0.0001. (**C**) Myofibril width is smaller in the absence of RSU1 but not FAK or tensin. (**D**) Sarcomere length is longer in the absence of RSU1 but unchanged in the absence of FAK or tensin. (**E**) MTZ length is longer in all three mutants; full quantitation is in *Figure 1—figure supplement 1*. We used a log plot in this and subsequent graphs measuring MTZ length due to the great range in length.

DOI: https://doi.org/10.7554/eLife.35783.003

The following figure supplement is available for figure 1:

**Figure supplement 1.** IAP mutant IFM defects are rescued by a copy of the IAP gene and phenocopied by RNAi in muscles.

DOI: https://doi.org/10.7554/eLife.35783.004

in muscle morphology were observed, but higher magnification revealed defects in myofibril morphology. To identify which phenotypes we should focus on, we first quantified aspects of myofibrils of flies that are 1 day old (after eclosion [AE]): myofibril width; sarcomere length; and the length of myofibril attachment sites, which in IFMs are called modified terminal Z-lines (MTZ) (*Reedy and Beall, 1993*) (*Figure 1B–D*). Myofibril width and sarcomere length were normal in IFMs lacking FAK or tensin, whereas IFMs lacking RSU1 had a reduced myofibril width and increased sarcomere length. Loss of each of these IAPs caused defects in MTZ length and unique alterations to MTZ structure (*Figure 1A*), so we focused our analysis on the MTZ. In wild type, the MTZ contains a brightly-staining block of actin, and the plasma membrane at the attachment site is interdigitated with the tendon cell, giving a wavy appearance (*Reedy and Beall, 1993*). The MTZ extends ~1 µm away from the plasma membrane to the Z-line at the start of the first sarcomere. In the absence of FAK the MTZ became modestly elongated, to 1.25 µm (*Figure 1A and E*). Myofibrils lacking RSU1 had an even more elongated MTZ, with a clear elongation of the interdigitations. Loss of tensin resulted in elongated and irregularly shaped MTZs, and approximately 20% of myofibrils were detached (*Figure 1A and E*). We confirmed that these defects were due to loss of these three IAPs, rather than any other mutation on the mutant chromosomes, as a transgenic copy of each gene rescued the defects (*Figure 1—figure supplement 1B and D*). In addition, these phenotypes result from loss of the IAP in the muscle since driving RNAi targeting these IAPs specifically in the muscles phenocopied the mutant phenotypes (*Figure 1—figure supplement 1B and D*).

We then used transmission electron microscopy (TEM) to examine the MTZ defects in more detail (*Figure 1B*). During the development of the muscles the attachment site becomes progressively more interdigitated with the tendon cell (*Reedy and Beall, 1993*), and this process appears defective in the absence of RSU1 and tensin. We see clear evidence of detachment of the muscle in the tensin mutant, but not in RSU1, showing that in the latter the more elongated MTZ does not reflect partial detachment. In the FAK mutant the electron dense layer adjacent to the plasma membrane, which we infer is the site of concentrated IAPs, was more continuous than in wild type (in wild type MTZs, this layer is reduced at the tips of interdigitations), thus revealing a role for FAK in keeping integrin adhesion structures as discrete contacts.

To discover when the defects first appeared, we next examined the MTZs at different stages of development. During pupal stages, the MTZs in all mutants appeared normal (*Figure 1A* and *Figure 1—figure supplement 1D*). However, immediately after eclosion, and prior to flight, all mutants showed an MTZ phenotype (*Figure 1A* and *Figure 1—figure supplement 1D*). This suggests that FAK, RSU1 and tensin are required either to build or strengthen the attachment so that it can withstand tension during pupal stages (*Weitkunat et al., 2014*). When we compared the phenotypes right after eclosion to 1 day later or 14 days later, we observed that the FAK mutant phenotype did not change, whereas the defects caused by absence of RSU1 or tensin became worse with age, presumably due to muscle activity. These defects were not strong enough to prevent flight (*Figure 1—figure supplement 1C*). Fewer flies lacking tensin and RSU1 flew than wildtype flies, but this defect

was not reproduced by muscle-specific knockdown of tensin or RSU1, suggesting that reduced flight is more likely caused by the wing blisters in these mutants.

As loss of tensin and RSU1 caused strong phenotypes, we hypothesised that their effects may be due to mislocalisation of integrins or other IAPs. We examined the distribution of integrins and other IAPs, using endogenous genes tagged with fluorescent proteins, all of which are fully functional (see Key Resources Table), but did not find any dramatic loss of those examined: βPS integrin, talin, PINCH, paxillin, GIT or vinculin (*Figure 2*, *Figure 2—figure supplement 1*). However, paxillin was elevated to 250% in the RSU1 mutant and to 150% in the tensin mutant. The IPP components ILK and PINCH were affected differently by the absence of RSU1; ILK was reduced 50% whereas PINCH was unaffected (*Figure 2L*). In addition, vinculin distribution showed the extreme stretching of the MTZ in the absence of tensin (*Figure 2F*). Loss of tensin did not alter RSU1 levels or distribution (*Figure 2H*), nor did loss of RSU1 alter tensin (*Figure 2K*). To summarize, we have discovered that removing each IAP caused a unique and relatively mild defect in myofibril attachment, none of which resulted in the loss of recruitment of another IAP.

## Genetic interactions reveal inhibitory action and compensation between IAPs

To test for compensatory activities between these proteins we examined the consequences of removing two at a time. Flies lacking RSU1 and tensin died at pupal stages, however in these pupae the MTZs appeared normal (*Figure 3—figure supplement 1A*), suggesting the lethality is caused by defects in another tissue. Therefore, we employed RNAi to specifically knockdown tensin in muscles, using the muscle specific driver mef2-Gal4. Loss of RSU1 did not worsen the phenotype of tensin knockdown (*Figure 3A* and *Figure 3—figure supplement 1C*), demonstrating that these two proteins do not compensate for each other.

Combining removal of FAK and RSU1 resulted in the unexpected rescue of the RSU1 phenotype (*Figure 3A*). This supports a role for FAK in downregulating integrin function (*Ilić et al., 1995*), and the expansion of integrin adhesions (*Figure 1B*). To confirm that increased integrin activity explains the rescue, we combined RSU1 loss with an integrin β subunit mutant with increased activity. We used the allele *mys[b28]*, a mutation in the hybrid domain (V423E) that strongly elevates affinity for ligand (*Kendall et al., 2011*), and this also rescued the phenotype (*Figure 3A* and *Figure 3—figure supplement 1C*). Activated integrin alone did not cause any detectable changes (*Figure 3A* and *Figure 3—figure supplement 1C*), allowing us to infer that the MTZ elongation caused by FAK loss is not due to elevating integrin, but rather loss of a positive contribution by FAK to MTZ formation. Consistent with this, activated integrin rescued MTZ elongation caused by the absence of FAK (*Figure 3A* and *Figure 3—figure supplement 1C*). This mutant combination also shows that 'double' elevation of integrin activity did not cause detectable defects. Thus, in IFMs, FAK has both positive and negative activities. Activated integrin also partially rescued tensin loss, but for this IAP removing FAK did not rescue (*Figure 3A* and *Figure 3—figure supplement 1C*). This suggests that tensin normally elevates integrin activity, and this is inhibited by FAK. This predicts that elevating FAK would inhibit tensin further, producing a phenotype similar to loss of tensin. Using the act88F-Gal4 driver to overexpress Fak-GFP, we observed a similar phenotype to loss of tensin (*Figure 3A* and *Figure 3—figure supplement 1C*). Of note, this is different than embryonic muscles, where FAK overexpression also causes muscle detachment, but loss of tensin does not (*Grabbe et al., 2004*; *Torgler et al., 2004*), and therefore demonstrates that FAK achieves negative regulation of integrins by different mechanisms in larval muscles and IFMs. Furthermore, the finding that activating integrin only partially rescues loss of tensin suggests that tensin does more than activate integrin; consistent with this, neither activated integrin nor loss of FAK rescued the wing blister phenotype of tensin mutants (data not shown).

We also examined overlap in function between these IAPs and talin, which is the single IAP to date that is required for all integrin adhesion in *Drosophila* (*Klapholz and Brown, 2017*). Talin reduction (by removing one gene copy) substantially enhanced the defects caused by tensin loss, but had no effect on FAK or RSU1 (*Figure 3A* and *Figure 3—figure supplement 1C*). Thus, full levels of talin are especially important in the absence of tensin, and the overlap in function between talin and tensin is consistent with both increasing integrin activity (*Georgiadou et al., 2017*; *Klapholz and Brown, 2017*).

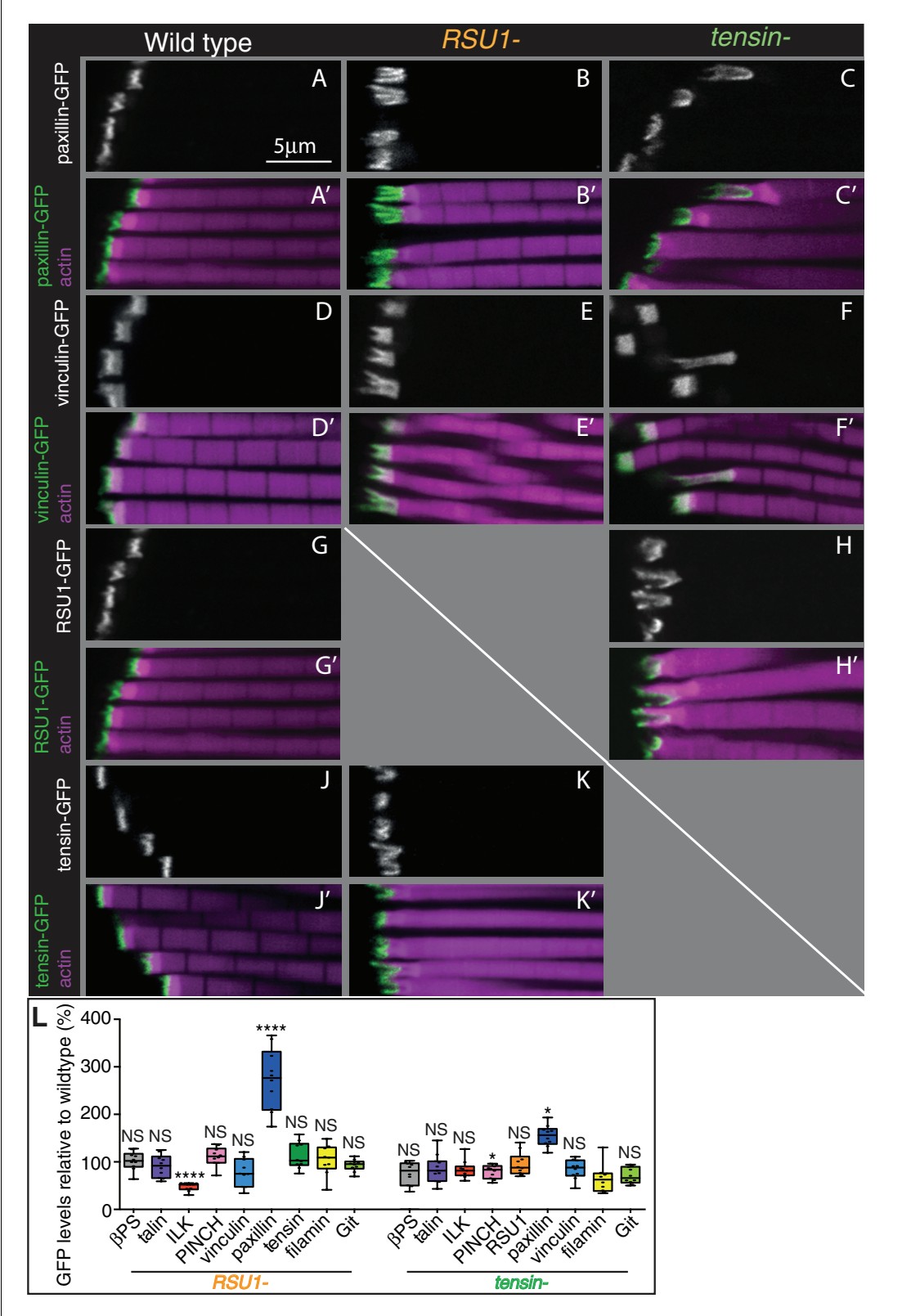

**Figure 2.** Other IAPs are still recruited in IFMs lacking RSU1 or tensin. (A–K') Confocal images showing the distribution of indicated GFP-tagged IAPs (white, green), each expressed from their own promoter, in the genotypes indicated at the top each panel. F-actin is magenta. The IFMs are from flies 1 day AE in this and all subsequent figures, unless indicated otherwise. (L) Quantification of IAP-GFP levels relative to levels in wild type IFMs at 1 day AE. For each genotype 10–30 myofibrils were measured from each of 10 individuals and an average from each individual was calculated; all data points are

*Figure 2 continued on next page*

*Figure 2 continued*

shown, overlain with a boxplot showing median, interquartile range, minimum and maximum; statistical test used was Mann-Whitney U. *p≤0.05; **p≤0.01; ***p≤0.001; ****p≤0.0001 .

DOI: https://doi.org/10.7554/eLife.35783.005

The following figure supplement is available for figure 2:

**Figure supplement 1.** Distribution of additional IAPs in IFMs lacking RSU1 or tensin.

DOI: https://doi.org/10.7554/eLife.35783.006

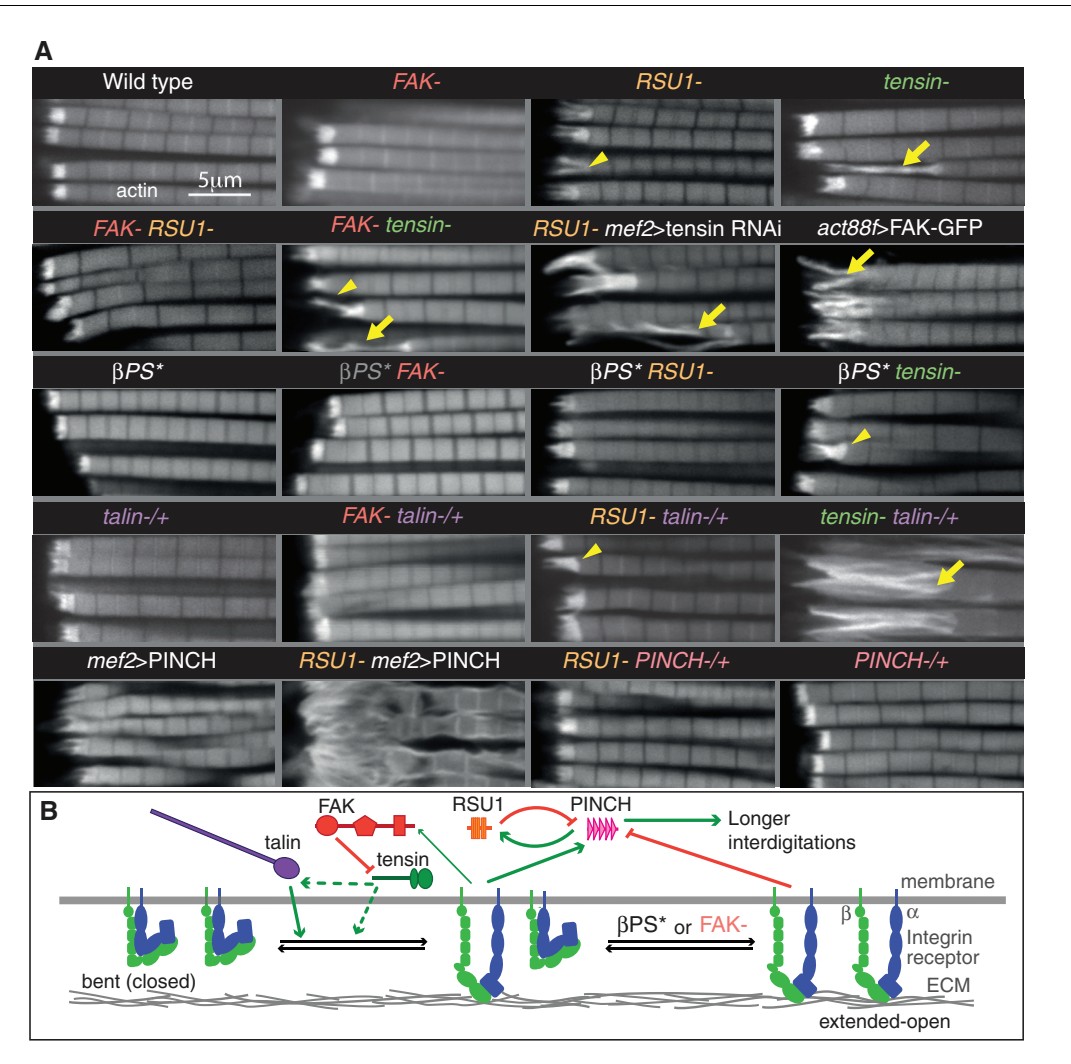

**Figure 3.** Epistasis reveals inhibitory action and compensation between IAPs. (A) F-actin in myofibrils of the indicated genotype. βPS* is a missense mutant that makes βPS more active (*mys[b27]*). Yellow arrowheads indicate myofibrils with elongated MTZs and yellow arrows detaching myofibrils. Quantitation of MTZ length is in *Figure 3—figure supplement 1*. (B) Model of IAP function. Integrin heterodimers (green and blue) exist in an inactive bent conformation and an open upright conformation that binds ECM. Talin (purple) and tensin (green) both promote integrin activation, and FAK (red) inhibits activation by tensin. Active integrin performs adhesive functions through IAPs, including FAK and RSU1 (orange). RSU1 promotes adhesion by inhibiting excessive PINCH (pink) activity. Constitutively active integrin (βPS*) or loss of FAK, increases the amount of active integrins and stimulates the function of other IAPs that compensate for the loss of RSU1 or FAK.

DOI: https://doi.org/10.7554/eLife.35783.007

The following figure supplement is available for figure 3:

**Figure supplement 1.** Defects are not enhanced in flies lacking both tensin and RSU1.

DOI: https://doi.org/10.7554/eLife.35783.008

We next explored the importance of RSU1 binding to PINCH for its function. Because loss of PINCH is embryonic lethal (*Clark et al., 2003*) and because the RSU1 mutant phenotype cannot be explained by a loss of PINCH recruitment (*Figure 2L*), we instead overexpressed PINCH, anticipating that this may rescue IFM defects since elevating PINCH levels partially rescues hypercontraction mutants in larval and flight muscle (*Pronovost et al., 2013*). Unexpectedly, driving high levels of PINCH alone with the Gal4 system was sufficient to cause defects closely resembling the loss of RSU1, and combining overexpression of PINCH with loss of RSU1 enhanced this phenotype (*Figure 3A* and *Figure 3—figure supplement 1C*). Conversely, reduction of PINCH (by removing one gene copy) partially rescued the loss of RSU1 (*Figure 3A* and *Figure 3—figure supplement 1C*). This indicates that the function of RSU1 is to inhibit PINCH activity, rather than aid its function. This led to a model (*Figure 3B*) where free PINCH has an important activity that must be carefully regulated and is suppressed by RSU1 binding or integrin activity (since activated integrin or loss of FAK rescues loss of RSU1). The wing blisters caused by the absence of RSU1 are also rescued by activating integrin or removing FAK (data not shown), suggesting that this phenotype is also caused by loss of PINCH inhibition. To further understand this PINCH activity we performed structure-function analysis using constructs which lacked either the RSU1-binding LIM5 domain and adjacent LIM4 domain (PINCHΔ4–5) or the ILK-binding LIM1 domain (PINCHΔ1) (*Figure 3—figure supplement 1*). PINCHΔ4–5 localized indistinguishably to full length PINCH and caused a similar, though weaker, phenotype, ruling out a model where RSU1 inhibits PINCH activity by competing off another protein that binds to LIM5 and is necessary for PINCH activity. PINCHΔ1 no longer caused a phenotype, nor was it tightly localized, showing that ILK-binding is essential for PINCH recruitment and activity, consistent with results in the larval muscles (*Zervas et al., 2011*). Overexpression of ILK did not cause a phenotype (*Figure 3—figure supplement 1*), suggesting that it is PINCH that is limiting, and supporting the importance of tightly regulating PINCH activity.

Taken together these findings have allowed us to propose a model for how these IAPs contribute to integrin function (*Figure 3B*). Both talin and tensin contribute to the activity of integrin, with tensin being inhibited by FAK to generate discrete integrin adhesive structures. RSU1 and integrin keep PINCH levels at the correct level; in the absence of RSU1 increased PINCH activity results in fewer, longer interdigitations. These findings highlight the tight balance in IAP activity required to make a wild type adhesive contact. With this improved understanding of the role of IAPs, we examined how vinculin fits into this picture.

## Vinculin has two distinct functions in the MTZ

The defects caused by FAK, RSU1 and tensin are all in the MTZ adhesion structure, close to the membrane. In contrast, loss of vinculin results in disruption to the actin organization up to 25 μm from the muscle ends, with a lack of a clear distinction between the MTZ and the terminal sarcomeres, although overall muscle morphology appeared normal (*Figure 4A*). This defect was rescued by a transgenic copy of vinculin or muscle muscle-specific expression of vinculin, confirming that the defect was caused by loss of vinculin (*Figure 4—figure supplement 1A and B*). The vinculin defect arose earlier than those caused by loss of FAK, tensin or RSU1, as an elongation of the terminal actin block was detectable in pupal stages (*Figure 4A* 48 hr APF and *Figure 4—figure supplement 1B*). The extent of actin disruption progressively increased as adult flies aged from 0 to 14 days (*Figure 4A* and *Figure 4—figure supplement 1B*). This was quantified by measuring the distance between muscle terminus and the first M-line, rather than MTZ length, because of the loss of a distinct edge of the MTZ. Analysis of the vinculin mutant by TEM (*Figure 4D*) showed defects in addition to the expansion of the MTZ. The even layer of electron dense material at the membrane appears disrupted on the myofibril side of the attachment, but not the tendon cell side. In some areas it is thinner, whereas other regions contain round electron-dense structures, which are also seen throughout the MTZ. One day old flies lacking vinculin were able to fly normally but after 14 days, only 20% were able to fly (*Figure 4B*). However, muscle specific knockdown of vinculin did not reduce flying ability, yet caused comparable IFM actin defects, indicating that loss of flight in flies lacking vinculin is not due to the IFM defects.

A role for vinculin in organizing actin distant from the membrane is consistent with the difference between the distribution of vinculin and other IAPs. Whereas the distribution of the other IAPs looked identical to integrin, vinculin extended up to 1 μm away from the muscle ends (*Figure 2D*), but not in the tendon cell, consistent with the greater importance of vinculin in maintaining an even

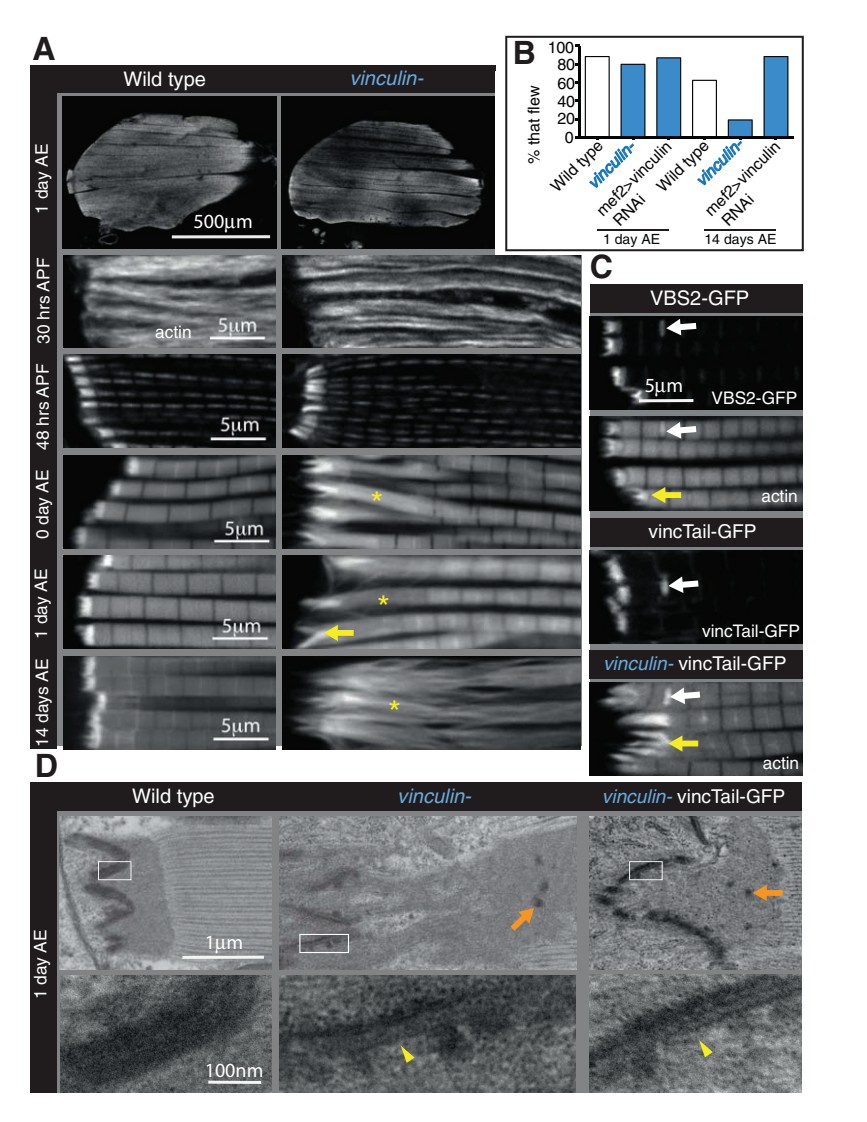

**Figure 4.** Integrin regulates actin structures at a distance, through vinculin. (**A**) F-actin distribution (white) in wild type and vinculin- IFMs at the stages indicated. Yellow asterisks and yellow arrow indicate regions of disorganized actin. (**B**) Muscle defects do not result in reduced flying ability in the vinculin mutant. 30 flies per genotype were tested and percentage that fly is shown. (**C**) F-actin distribution (white) in IFMs of the genotype indicated. Vinculin tail and VBS2 were expressed with *mef2*-Gal4. White arrows indicate zebra bodies (abnormally large Z-lines). Yellow indicate detaching myofibrils. (**D**) TEM images of MTZs at the indicated stage and genotype. Orange arrows indicate electron dense aggregates in the MTZ. Yellow arrowheads indicate loss of electron dense material from the muscle, but not tendon cell, membrane.

DOI: https://doi.org/10.7554/eLife.35783.009

The following figure supplement is available for figure 4:

**Figure supplement 1.** Defects seen with loss of vinculin are rescued by adding back wild type vinculin.

DOI: https://doi.org/10.7554/eLife.35783.010

layer electron dense material on the myofibril side of the attachment. We confirmed this with increased resolution microscopy, using structured illumination microscopy (SIM) (*Figure 5*). Whereas tensin and RSU1 colocalized with integrin, vinculin was separate from integrin (*Figure 5C–G*). GFP on the N-terminus of talin also colocalised with integrin, whereas a more C-terminal insertion of GFP into talin extended 0.5–1 µm from the membrane (*Figure 5B*). Thus, it is feasible for the distal

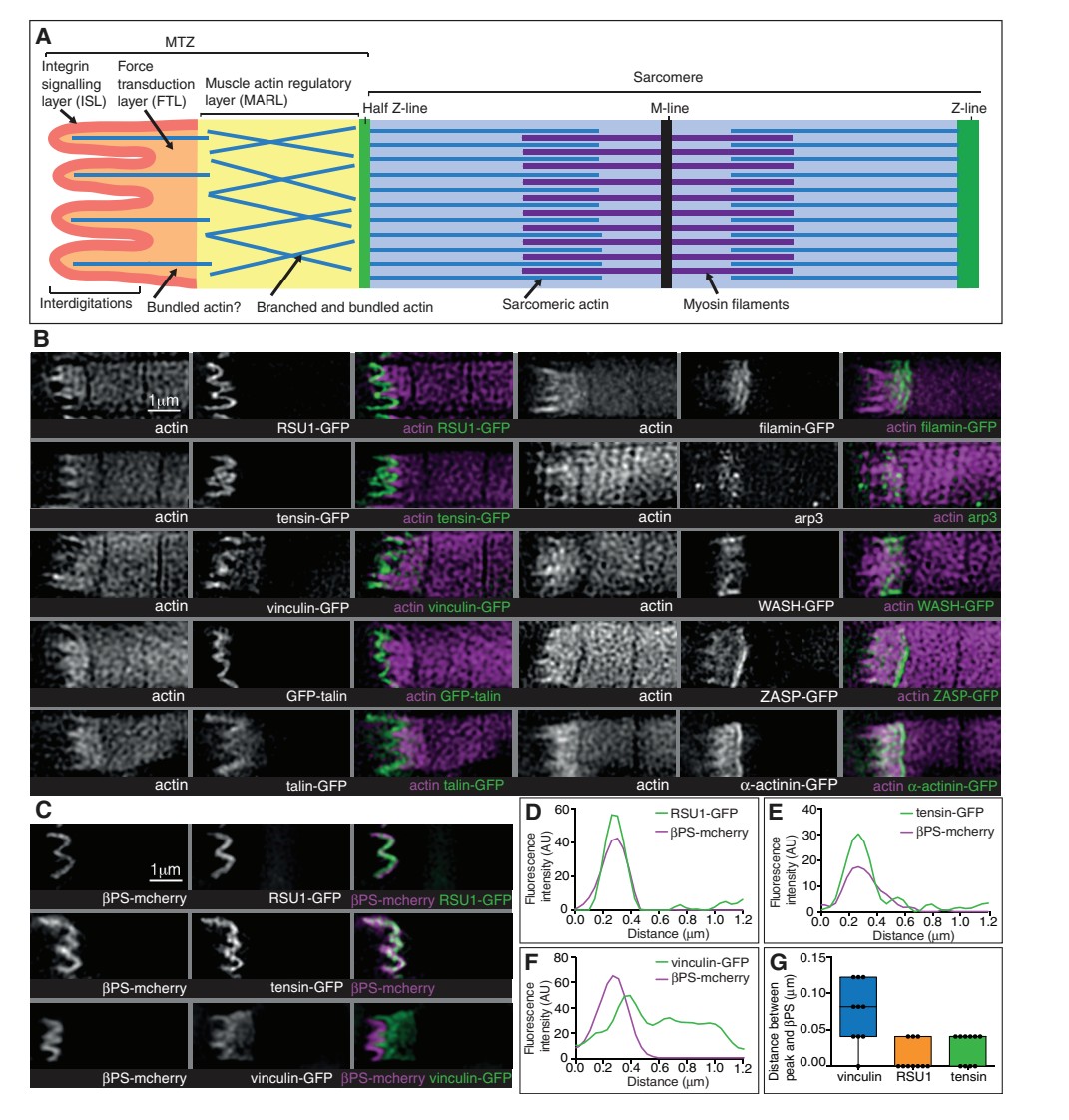

**Figure 5.** Super-resolution microscopy resolves distinct zones in the MTZ. (**A**) Schematic of the four zones within the MTZ: (1) an integrin signalling layer (ISL, red) at the membrane; and then zones containing different actin structures-(2) a force transduction layer (FTL, orange); (3) a muscle actin regulatory layer (MARL, yellow); and (4) the first (half) Z-line (green) followed by the first sarcomere (blue). (**B**) SIM images of wild type myofibrils expressing IAPs tagged with GFP as indicated, or using antibody staining for arp3 (white and green), and stained for F-actin (white and magenta). Note the four zones: (1) the ISL containing tight membrane localization of RSU1, tensin, and GFP-talin tagged at the N-terminus and very little F-actin; (2) both ISL and FTL for vinculin and talin tagged toward the C-terminus; (3) the MARL containing filamin, vinculin, arp3 and WASH; (4) the first Z-line containing high levels of ZASP and α-actinin. Note that the web-like appearance of actin in the SIM images is an artefact of the SIM technique. See also *Figure 5—figure supplement 1*. (**C–F**) RSU1 and tensin overlap with integrin, in the ISL, whereas vinculin is separate, within the FTL. (**C**) SIM images of wild type myofibrils expressing pairs of proteins as indicated. (**D–F**) Representative line graphs, from images as in B, showing fluorescence intensity over distance. (**G**) Quantitation of the distance between the peaks of fluorescent intensity of βPS-mcherry versus RSU1-GFP, tensin-GFP or vinculin-GFP, combined from measurements of 10 myofibrils each. Boxplots show median, interquartile range, minimum and maximum.

DOI: https://doi.org/10.7554/eLife.35783.011

The following figure supplement is available for figure 5:

**Figure supplement 1.** Comparison of IAP distribution at the MTZ versus Z-lines.

DOI: https://doi.org/10.7554/eLife.35783.012

accumulation of vinculin to occur by binding stretched talin. This fits with the stretching of talin in fibroblasts of ~500 nm (*Margadant et al., 2011*), and additional stretching of vinculin.

With further experimental perturbation, we were able to divide the defects caused by loss of vinculin into two components, one perturbing adhesion, similar to the defects caused by FAK, RSU1 and tensin, and the second disrupting actin organization. The division was achieved by eliminating the binding of vinculin head to talin, either by deletion of the head domain, or by competitive inhibition through overexpression of a single vinculin-binding-site helix from talin (*Maartens et al., 2016*). Both caused aberrant attachment, but little disruption to actin in adjacent sarcomeres (*Figure 4C* and *Figure 4—figure supplement 1B*). In addition, TEM imaging revealed that vinculin tail did not rescue the disruption to electron dense material at the membrane, nor the appearance of aggregates away from the membrane (*Figure 4D*). Thus, vinculin binding to talin contributes to integrin-mediated adhesion, and surprisingly the tail domain could function on its own to keep actin organized. To our knowledge this is the first demonstration that the tail domain can function independently, and it is also significant that the tail alone gets recruited to the MTZ (*Figure 4C*).

## Vinculin helps organize a novel actin structure at the muscle termini

It was difficult to discern how the extended actin defect arose in the absence of vinculin, as it was not clear which actin structure became disorganized, whether: (1) MTZ actin expands, pushing the first sarcomere away; (2) MTZ actin mixes with sarcomeric actin; or (3) overall actin organization near muscle ends deteriorates. To resolve between these possibilities, we examined actin-binding proteins and components of muscle sarcomeres, which revealed three major insights.

First, the distribution of the M-line component Unc-89/obscurin ruled out model 1 of MTZ pushing the first sarcomere away, as M-lines were visible within the region of disrupted actin (*Figure 6H*). Second, using super-resolution microscopy (*Figure 5*) and standard confocal microscopy (*Figure 5—figure supplement 1*) we were able to divide the MTZ into four distinct regions based on their composition, which appear analogous to the different regions discovered by super-resolution in the Z-axis of focal adhesions (*Kanchanawong et al., 2010*). The four regions are: (1) the adhesion structure at the membrane, analogous to the integrin signalling layer (ISL) in focal adhesions, which contains most IAPs (N-terminus of talin, paxillin, GIT, ILK, PINCH, RSU1, FAK and tensin); (2) the next layer, analogous to the force transduction layer (FTL), containing actin, the C-terminus of talin and vinculin; (3) an unexpected novel, actin-rich region, which appears analogous to the actin regulatory layer, but instead of containing Ena/VASP and zyxin, contains filamin, Arp3, and the Arp2/3 activator WASH, as well as the C-terminus of talin and vinculin. Because of these differences, we term this novel layer the <u>m</u>uscle <u>a</u>ctin <u>r</u>egulatory <u>l</u>ayer or MARL; and (4) the first Z-line with high levels of ZASP and α-actinin, analagous to the stress fiber attached to the focal adhesion (*Figure 5A and B*). The segregation of these proteins is not absolute, as ZASP and α-actinin were present in both the FTL and MARL, and in addition components from the MARL were detected at low levels in all Z-lines (*Figure 5—figure supplement 1*).

The third key finding is that the MARL is expanded in the absence of vinculin (e.g. filamin, ZASP), whereas the ISL (e.g. paxillin) was not substantially altered (*Figures 6B,D,F,J* and *Figure 6—figure supplement 1B*). In addition, filamin levels were reduced to 40% in the absence of vinculin (*Figure 6J*). Thus, these findings favour explanation 2, where it is the MARL that expands into the sarcomeres, and therefore that vinculin has a role in the formation and/or stability of the MARL, consistent with the function of the tail domain in this region.

The MARL is distinguished by two actin-binding proteins associated with branched actin networks, Arp2/3 and filamin. Actin within the FTL is likely to link the MARL to the membrane, whereas the first Z-line assembles at the outer edge of the MARL, where it anchors the parallel actin filaments of the sarcomere. This highlights the ability of actin-binding proteins to associate with discrete actin structures in adjacent parts of the cell. Vertebrate muscles contain filamin-C at the myotendinous junction (*van der Ven et al., 2000*), suggesting they have a comparable structure. Our next goal was to discover how vinculin contributes to the formation and function of the MARL.

## Filamin and WASH contribute to MARL formation

The concentration of Arp2/3, its nucleation promoting factor WASH, and filamin in the MARL suggested that the actin in this structure may form a cross-linked network, and we sought to determine

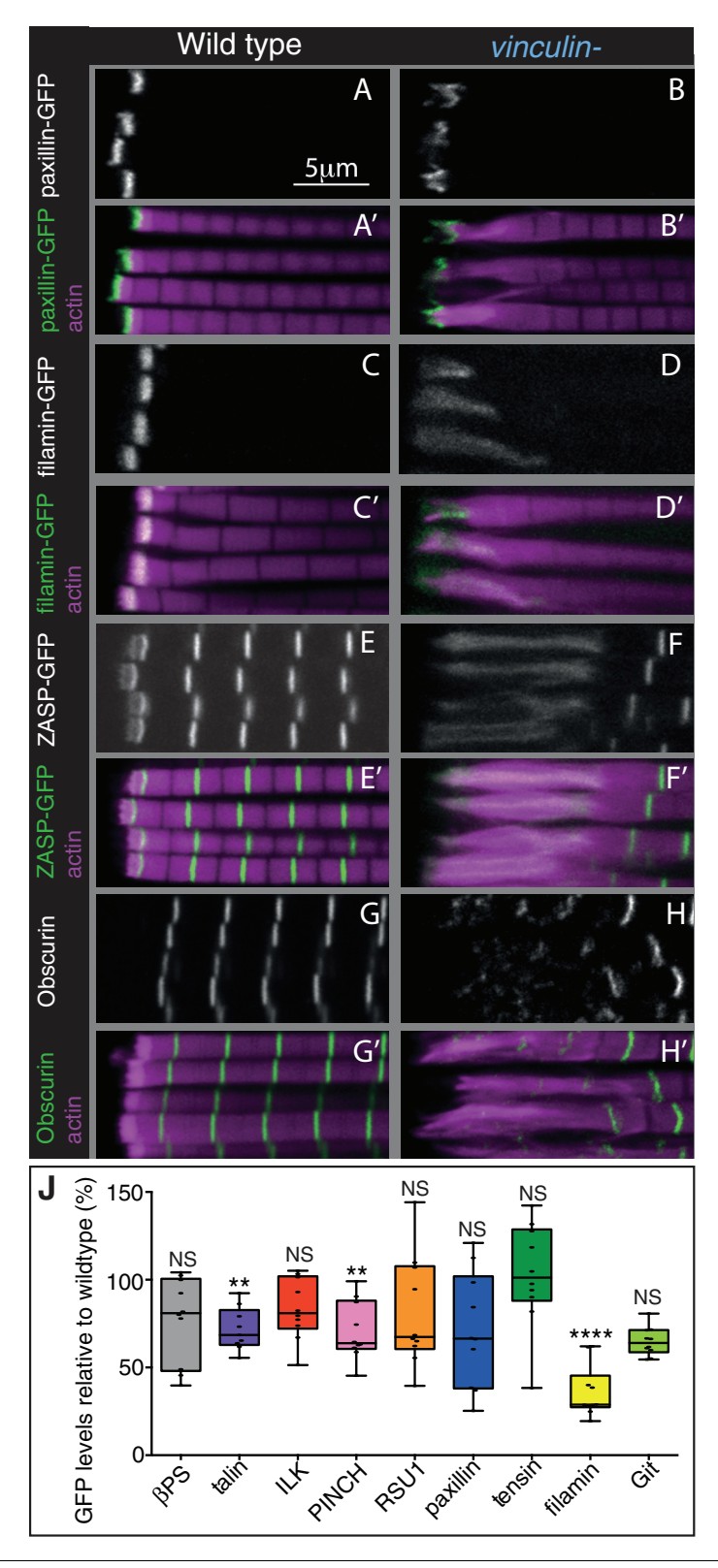

**Figure 6.** The absence of vinculin causes expansion of the MARL into the sarcomeres. (**A–H'**) Distribution of F-actin (magenta) and the protein indicated (white, green) in wild type and vinculin- IFMs. MARL expansion is seen by expanded filamin-GFP and ZASP-GFP, and the underlying sarcomeres are visible with the M-line protein Obscurin/Unc-89 (antibody staining), whereas the ISL seems unaffected, as marked with paxillin-GFP. (**J**) *Figure 6 continued on next page*

*Figure 6 continued*

Quantification of IAP-GFP levels relative to levels in wild type IFMs at 1 day AE. For each genotype 10–30 myofibrils were measured from each of 10 individuals and an average from each individual was calculated; all data points are shown, overlain with a boxplot showing median, interquartile range, minimum and maximum; statistical test used was Mann-Whitney U. *$p \leq 0.05$; **$p \leq 0.01$; ***$p \leq 0.001$; ****$p \leq 0.0001$.

DOI: https://doi.org/10.7554/eLife.35783.013

The following figure supplement is available for figure 6:

**Figure supplement 1.** Distribution of additional IAPs in IFMs lacking vinculin.

DOI: https://doi.org/10.7554/eLife.35783.014

whether these proteins contribute to MARL actin assembly. Flies lacking filamin are viable, albeit female sterile due to defects in the actin-based ring canals during oogenesis (*Robinson et al., 1997*). Loss of filamin (using a deletion of the C-terminal half [*Huelsmann et al., 2016*]) did not alter overall muscle shape, but caused reduced MARL size in pupal stages and just after eclosion (*Figure 7A* and *Figure 7—figure supplement 1B and D*), which converted to MARL expansion 1 day after eclosion and thereafter (*Figure 7A* and *Figure 7—figure supplement 1B*), resembling the vinculin mutant phenotype. The reduction in the MARL is even more dramatic when visualized by TEM (*Figure 7B*). A flight defect was observed in flies lacking filamin overall and when specifically knocked down in muscles (*Figure 7C*), consistent with previous observations (*González-Morales et al., 2017*).

Muscle-specific knockdown of Arp2/3 components is lethal and results in severe muscle defects (*Schnorrer et al., 2010*). However, flies lacking the Arp2/3 nucleation promotion factor WASH are viable (*Nagel et al., 2017*), and we found they had no gross changes to muscle morphology, had a smaller MARL during pupal stages, and a progressively elongated MARL during adult stages, similar to loss of filamin (*Figure 7A* and *Figure 7—figure supplement 1B and D*). The F-actin staining in the MARL was reduced (*Figure 7A*) in the absence of WASH, as were levels of filamin (*Figure 7—figure supplement 1E and F*). TEM confirmed that the MARL was less dense in MTZs lacking WASH and showed defects in the morphology of the interdigitations (*Figure 7B*). Muscle-specific RNAi of filamin and WASH confirmed that this phenotype is due to loss of these proteins in the muscles (*Figure 7—figure supplement 1A and B*). In contrast to filamin, flies lacking WASH flew normally (*Figure 7C*), suggesting it is the defects in sarcomere structure (*González-Morales et al., 2017*) rather than defects at the MTZ that impair flight in the absence of filamin.

We were intrigued by the observation that two proteins that aid MARL formation, vinculin and filamin, are mechanosensitive (*Rognoni et al., 2012*; *Atherton et al., 2016*; *Huelsmann et al., 2016*). We therefore wondered whether mechanical signaling might be required to build the MARL. To test the role of the mechanosensing region of filamin, we examined the IFM defects caused by mutations that either delete it (ΔMSR), make it harder to open (closed), or make it easier to open (open) (*Huelsmann et al., 2016*). The ΔMSR and closed mutant had similar defects as deletion of the C-terminus (*Figure 7D*), although they were not as strong, indicating that mechanosensing contributes the majority, but not all, of filamin activity in forming the MARL. The open filamin did not cause any MTZ defects, but we observed a gain of function phenotype, consisting of the formation of bright, ectopic bands of actin, usually associated with a Z-line (*Figure 7D*). Similar ectopic actin bands, termed 'zebra bodies', occur with missense mutations in the troponin gene (*Nongthomba et al., 2007*) or knockdown of muscle components (*Schnorrer et al., 2010*). Overexpression of wild type vinculin or vinculin tail also produced zebra bodies, usually at the second-most terminal Z-line (*Figure 7—figure supplement 1G* and *Figure 4B*). Overexpression of WASH was sufficient to enlarge the MARL, with an apparent increase in actin levels (*Figure 8A* and *Figure 7—figure supplement 1B*).

We further manipulated MARL proteins to determine how they worked together. None of the double mutant combinations of vinculin, filamin and WASH showed enhancement or suppression of the expected additive phenotype (*Figure 8A* and *Figure 7—figure supplement 1B and C*). As these are all null alleles, this shows they are all required in the same pathway to make the MARL. It is also of interest that the reduced MARL in the absence of WASH does not result in a reduction in the expanded actin caused by loss of vinculin or filamin. Open filamin partially rescued the loss of vinculin (*Figure 8A* and *Figure 7—figure supplement 1C*), showing that a key function of vinculin is to

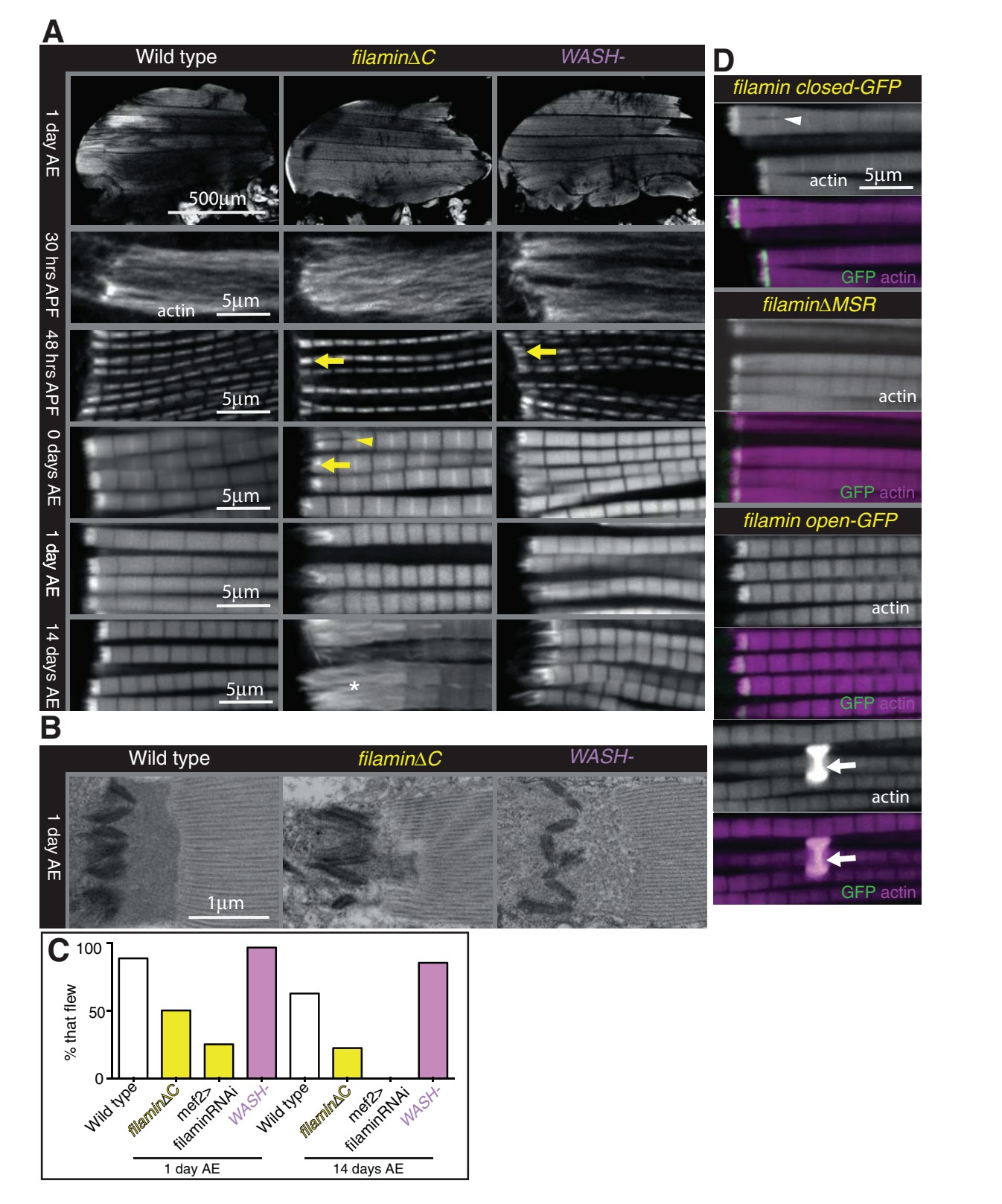

**Figure 7.** Filamin and WASH contribute to MARL formation. (**A**) Loss of filamin or WASH results in a smaller MARL at 48 hr APF and 0 days AE, followed by an expansion of the MARL at later stages. F-actin distribution (white) of the genotypes and stages as indicated. Yellow arrows indicate small MTZs, asterisks indicate expanded and disorganised MARL, and yellow arrowheads indicate split myofibrils observed in these mutants but not vinculin

*Figure 7 continued on next page*

*Figure 7 continued*

(*Figure 4*). (B) TEM images of MTZs at the indicated stage and genotype. (C) Muscle defects caused by loss of filamin, but not WASH, lead to reduced flying ability. 30 flies per genotype were tested and percentage that fly is shown. (D) Mechanical opening of filamin is required for MARL formation. F-actin and filamin-GFP distribution. Filamin that is easier to open makes a normal MTZ, and ectopic zebra bodies (white arrows). White arrowheadindicates split myofibrils.

DOI: https://doi.org/10.7554/eLife.35783.015

The following figure supplement is available for figure 7:

**Figure supplement 1.** Filamin and WASH contribute to MARL formation.

DOI: https://doi.org/10.7554/eLife.35783.016

open filamin. In contrast, open filamin did not rescue the absence of WASH, suggesting that WASH is downstream of filamin. Overexpression of WASH did not rescue the loss of vinculin or filamin and instead made the loss of filamin phenotype worse (*Figure 8A* and *Figure 7—figure supplement B and C*); reciprocally, filamin does not appear to be required for overexpressed WASH to expand the MARL, judging by the elevated actin staining at the termini of the disrupted actin. Thus, the aberrant MARL that forms in the absence of filamin appears expanded by additional WASH. Consistent with filamin being downstream of vinculin, vinculin tail-induced zebra bodies contained filamin, and did not form when only closed filamin was present (*Figure 7—figure supplement 1G*), whereas open filamin still made zebra bodies in the absence of vinculin (*Figure 7—figure supplement 1G*).

The genetic interactions suggested that vinculin tail and filamin may interact directly or indirectly. To test this, we targeted a portion of filamin to the surface of the mitochondria with an ActA peptide (*Bubeck et al., 1997*) to test whether it could recruit vinculin tail within IFMs. We tagged the short version of filamin (filamin90, containing the C-terminal 9 filamin repeats, including the 6 that form the MSR and dimerization repeat), both in open and closed form, with RFP and the ActA mitochondrial targeting sequence. When expressed in the IFMs, the majority of the fusion protein formed clusters on the mitochondrial surface, but some was still recruited to the MARL (*Figure 8B*). All of the clusters formed by open filamin90 recruited large amounts of actin, whereas only a fraction of the closed filamin90 clusters did so (*Figure 8B*). Both were able to recruit vinculin tail (*Figure 8C*). Furthermore, the recruitment of these fusion proteins to the MARL required vinculin (*Figure 8E*). Overexpression of WASH resulted enlarged the clusters formed by both closed and open filamin 90 (*Figure 8D* and not shown), with the filamin surrounding a sphere of actin and WASH-GFP. Thus, the C-terminus of filamin is able to recruit vinculin, actin and WASH to an ectopic location.

We can put these results together into a speculative model (*Figure 8F*) whereby formation of the MARL involves a series of mechanotransduction events: the first step is the mechanical stretching of talin, which will stabilize the open conformation of vinculin and expose the actin-binding tail; in the second step, the vinculin tail anchors the C-terminus of filamin to actin filaments, thus generating three actin contact points for the filamin dimer and permitting it to sense stretch in a more dense actin meshwork; in the third step, forces on the actin meshwork linked by filamin and vinculin leads to opening up of the filamin MSR, which then activates WASH, promoting Arp2/3-mediated formation of new actin branches, thus expanding the MARL.

## Discussion

The adult indirect flight muscles of *Drosophila* have proved to be an excellent system to identify functions for integrin-associated proteins (IAPs) that are not essential for viability. The mechanical linkage between the last Z-line of each myofibril and the plasma membrane is a well ordered and multi-layered structure, ideal for elucidating the mechanisms by which actin can be organized into different structures at subcellular resolution. In the layer closest to the membrane, the integrin signaling layer, we find important counterbalancing between IAPs, with FAK inhibiting the activation of integrin by tensin, and RSU1 inhibiting excess PINCH activity(summarised in *Figure 9*). We discovered that the muscle actin regulatory layer (MARL) has a different composition to the fibroblast ARL, containing a mechanotransduction cascade of vinculin and filamin, which, together with WASH and the Arp2/3 complex, builds an actin-rich zone linking the adhesion machinery at the membrane to the first Z-line (*Figure 9*).

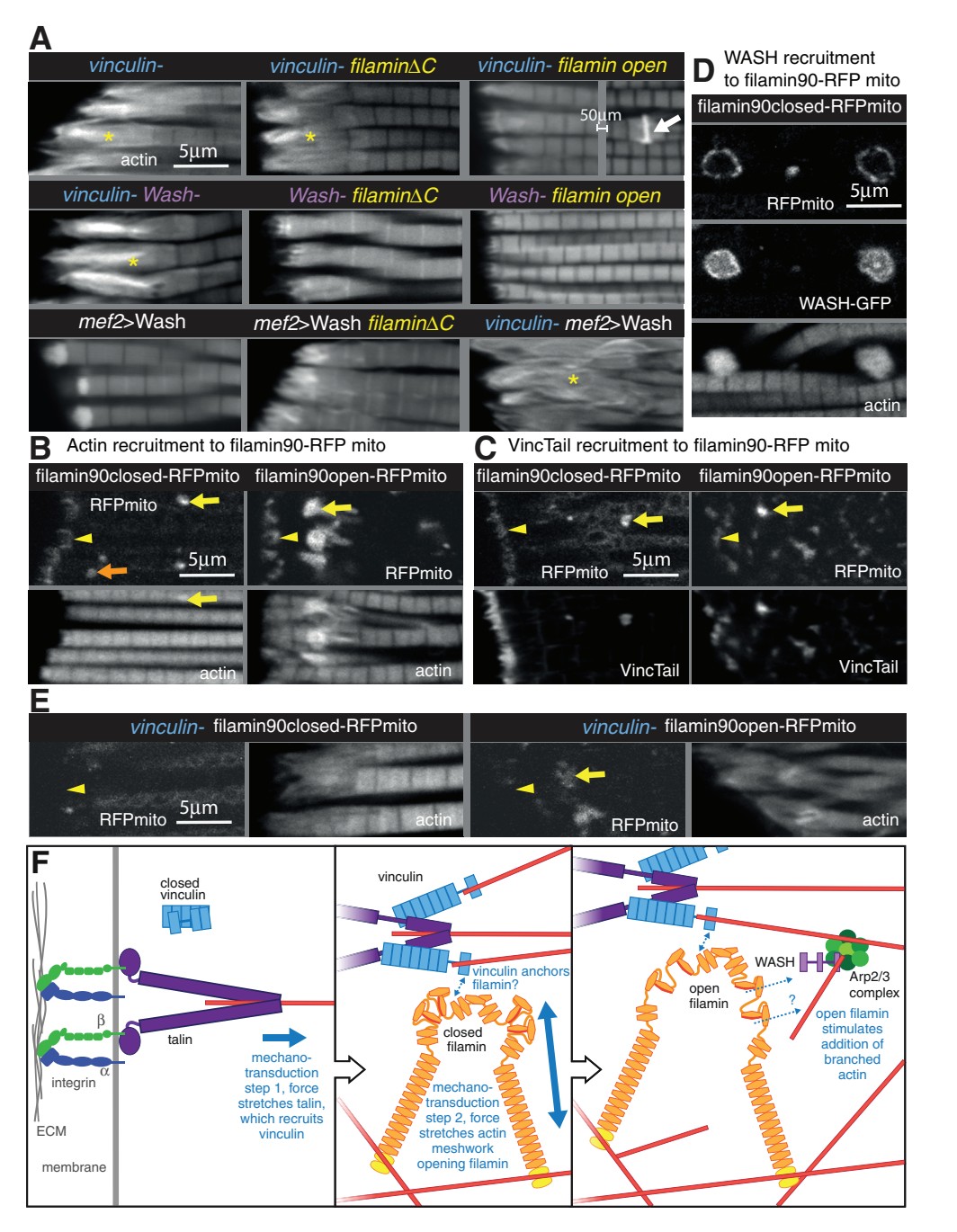

**Figure 8.** Vinculin, filamin and WASH work together to build the MARL. (**A**) F-actin distribution of the genotypes indicated. Yellow asterisks indicate disorganised actin. Combining mutants does not enhance the phenotypes, filamin open rescues absence of vinculin but not WASH, and overexpression of WASH expands the MTZ and enhances loss of filamin. White arrow indicates zebra bodies. (**B**) Filamin90open recruits actin more robustly than filamin90closed. Yellow arrows indicate mitochondrial aggregates that rcruit actin, orange arrows indicate aggregates that do not recruit actin and yellow arrowheads indicate the MTZ. (**C**) Filamin90open and closed-RFPmito recruit vinculinTail. (**D**) Filamin90closed recruits WASH-GFP, leading to large spheres of F-actin. (**E**) Vinculin is required for recruitment of filamin90-RFPmito constructs to the MTZ. (**F**) Model of MARL formation by vinculin (blue), filamin (yellow) and WASH (lilac). Mechanical stretching of talin (purple) reveals cryptic binding sites for vinculin. Vinculin binding to talin opens up vinculin, revealing it's actin binding tail. The tail anchors filamin to actin filaments causing the MSR of filamin to be stretched. Open filamin then stimulates addition of branched actin filaments through WASH and Arp2/3 (green) activity.

DOI: https://doi.org/10.7554/eLife.35783.017

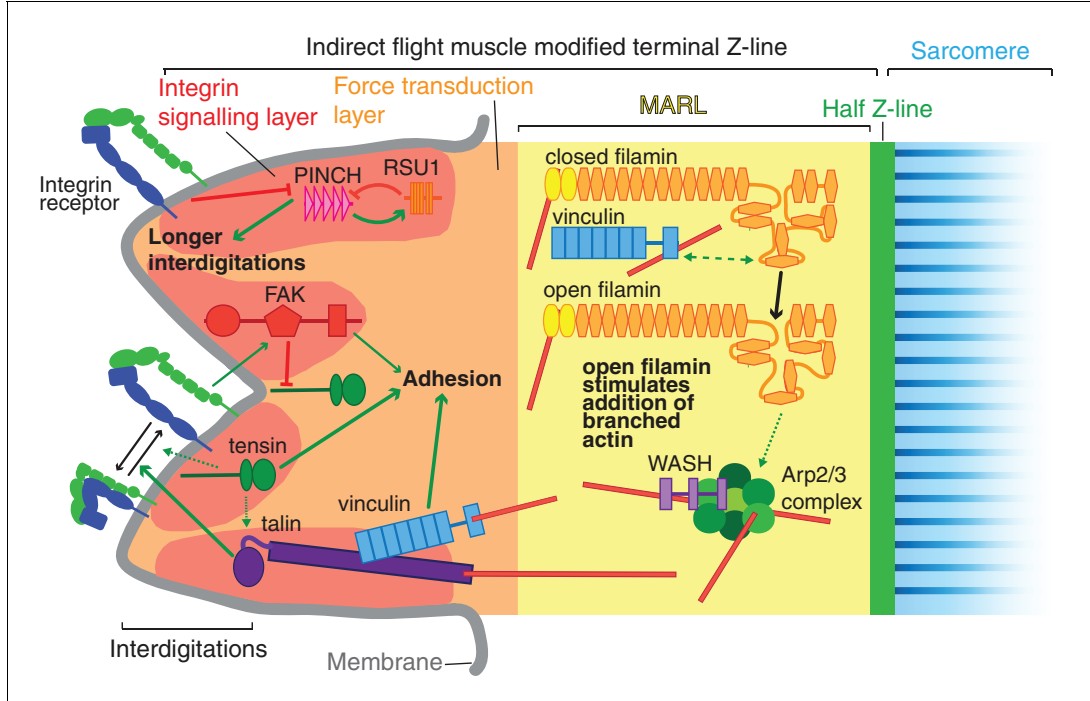

**Figure 9.** Model of IAP function in the IFM MTZ. The MTZ is composed of 4 zones: (1) an integrin signalling layer (ISL, red) at the membrane; and then zones containing different actin structures-(2) a force transduction layer (FTL, orange); (3) a muscle actin regulatory layer (MARL, yellow); and (4) the first (half) Z-line (green) followed by the first sarcomere (blue). Within the ISL, RSU1 (orange), PINCH (pink), tensin (green) and FAK (red) regulate integrin adhesion. Tensin promotes integrin activation, possibly through talin, and this activity is inhibited by FAK. Excessive PINCH activity, which leads to longer interdigitations than wildtype, is inhibited by RSU1. Within the FTL, stretched talin recruits and opens up vinculin, revealing its actin binding tail. In the MARL, vinculin tail anchors filamin to actin, leading to stretching of filamin and opening up of its mechanosensor region. Open filamin promotes addition of branched actin, possible through activation of WASH and the Arp2/3 complex.

DOI: https://doi.org/10.7554/eLife.35783.018

## FAK inhibits integrins via tensin

The MTZ revealed both positive and inhibitory actions of FAK, with the latter consistent with the role of FAK in adhesion disassembly (*Ilić et al., 1995*). Both loss of FAK and activated integrin supressed the phenotypes caused by loss of RSU1 or vinculin, but only activated integrin alleviated the defects caused by the absence of tensin, suggesting that FAK inhibition requires tensin activity, and in turn, tensin elevates integrin activity. This fits with the recent discovery that tensin contributes to the inside-out activation of integrins via talin (*Georgiadou et al., 2017*). FAK and tensin thus form a balanced cassette that we imagine responds to upstream signals to regulate integrin activity. Further work is needed to discover how tensin increases integrin activity, how this is inhibited by FAK, and what signals control this regulatory cassette. One model would have tensin activating integrin by direct binding to the β subunit cytoplasmic tail, and FAK inhibition by phosphorylation of tensin, but an alternative is that they have antagonistic roles in integrin recycling (e.g. *Rainero et al., 2015*).

## RSU1 keeps PINCH activity in check

RSU1 is part of the complex containing ILK, PINCH and Parvin (IPP complex), and binds the 5th LIM domain of PINCH (*Kadrmas et al., 2004*). Loss of RSU1 causes milder phenotypes than loss of ILK, PINCH or parvin, and these phenotypes have previously been interpreted as a partial loss of IPP activity. Our findings indicate that the phenotypes observed in the absence of RSU1 are due to too much PINCH activity, and therefore the role of RSU1 is to keep PINCH activity in check. This suggests that PINCH is perhaps the key player of the IPP complex, and is recruited to adhesions by integrin via ILK, and kept in check by integrin and RSU1. The importance of regulating active PINCH levels is consistent with the dosage sensitivity of PINCH: reducing PINCH partially rescues the dorsal closure defect in embryos lacking the MAPK Misshapen (*Kadrmas et al., 2004*), and elevating

PINCH rescues hypercontraction caused by loss of Myosin II phosphatase (*Pronovost et al., 2013*). Reducing the interaction of PINCH with ILK had unexpectedly no phenotype, but in combination with the loss of RSU1 becomes lethal (*Elias et al., 2012*); the lethality can now be interpreted as being caused by too much PINCH activity, rather than too little. Excess 'free' PINCH results in elongated membrane interdigitations and elevated paxillin levels. This suggests that PINCH has an important role at the cell cortex, consistent with cortical proteins in the PINCH interactome (*Karaköse et al., 2015*). Too much parvin activity also causes lethality, which is suppressed by elevating ILK levels (*Chountala et al., 2012*). Thus, it is increasingly clear that the functions of IPP components need to be tightly controlled. We gained some insight into how RSU1 inhibits PINCH activity by demonstrating that ΔLIM4, 5 PINCH still caused longer interdigitations. This rules out RSU1 blocking the binding of another protein from binding LIM5, and suggests instead that RSU1 bound to LIM5 must be inhibiting the activity of LIM1-3.

## Vinculin, Filamin and WASH contribute to the formation of the MARL

Vinculin has a dual function in the MTZ: its head domain promotes FTL stability via binding talin, and its tail promotes MARL formation. Our analysis of the vinculin mutant by electron microscopy showed a phenotype within the electron dense layer close to the membrane that we presume corresponds to the integrin signalling layer. It suggests that vinculin may mediate interactions between IAPs that aid in keeping this as an even layer. The fact that the disruption to this layer is only evident on the muscle side of the interaction raises the question of how similar the integrin junctions are on the two sides of this cell-cell interaction via an intervening ECM. Many other sites of integrin-mediated adhesion in *Drosophila* involve integrins on both sides of the interaction (*Bökel and Brown, 2002*) and by electron microscopy the electron dense material looks similar on the two sides (*Reedy and Beall, 1993*), and we would expect that both sides need to resist the same forces. Even with structured illumination microscopy we cannot resolve the two sides of the membrane, but our results show that the C-terminus of talin and vinculin are not pulled away from the membrane in the adult tendon cells. This suggests either that vinculin has a different role in the tendon cell, with a different configuration, as we observed for talin in the pupal wing (*Klapholz et al., 2015*), or it is absent.

The vinculin tail function in MARL formation does not require that vinculin is bound to talin, but we suspect that in the wild type it is talin-binding that converts vinculin into an open conformation, permitting the tail to trigger MARL formation with filamin, as outlined in our working model in *Figure 8F*. A key function of vinculin tail in the MARL is to aid the mechanical opening of the filamin mechanosensitive region. We presented evidence suggesting this is achieved by the vinculin tail anchoring the C-terminus to actin, but further work is required to determine if there is direct binding between the two proteins. Similarly, our results indicate that the Arp2/3 nucleation promoting factor WASH is part of the same pathway as filamin and acts downstream of it, but the connection between the two has yet to be resolved. This new function for WASH is distinct from its best characterized role regulating actin on intracellular vesicles during endosomal sorting and recycling (*Nagel et al., 2017*), but WASH also has additional roles in the nucleus and the oocyte cortex (*Verboon et al., 2018*, *2015*), showing that it is a versatile protein.

## Implications for muscle function and disease

Given the myofibril defects seen with loss of RSU1, tensin, vinculin and filamin it might be expected that mutations in genes encoding these IAPs might be implicated in muscle disease. Indeed, mutations in integrin α7, talin and ILK are associated with muscular myopathies in humans and mice (*Winograd-Katz et al., 2014*). Mutations in the genes encoding RSU1, tensin and vinculin have not been linked to muscle myopathies, but mutations in filamin are linked to myofibrillar myopathies (*Vorgerd et al., 2005*). However, given the subtlety of these defects in *Drosophila*, one might predict that mutations in genes encoding these IAPs are associated with subtle defects in humans such as reduced sporting performance or susceptibility to muscle injury. We are not aware of any mutations in genes encoding these IAPs being related to athletic performance or injury susceptibility (*Maffulli et al., 2013*), but we expect that these IAPs would be good candidates for further study in this area.

One way that these IAPs may contribute to athletic performance is by building a muscle shock absorber, the MARL, which protects the myofibrils from contraction-induced damage. The concept of muscle shock absorbers is well established since tendons perform this function (*Roberts and Azizi, 2010*). The presence of filamin, Arp3, vinculin and α-actinin in the MARL suggests that the MARL contains branched and bundled actin filaments. Branched actin networks have been shown to be viscoelastic (*Blanchoin et al., 2014*) and actin crosslinkers such as filamin have been shown to reduce viscosity and increase elasticity of actin networks (*Koenderink et al., 2009*). Further study into the functional nature of the MARL should increase our understanding of athletic performance and injury susceptibility.

# Materials and methods

**Key resources table**

| Reagent type (species) or Resource | Designation | Source or reference | Identifiers | Additional information |
|---|---|---|---|---|
| Antibody | rabbit anti-obscurin | (*Burkart et al., 2007*) | N/A | |
| Antibody | rabbit anti-arp3 | (*Stevenson et al., 2002*) | N/A | |
| Antibody | rat anti-filamin C terminus | (*Sokol and Cooley, 1999*) | N/A | |
| Genetic reagent (*Drosophila melanogaster*) | FAK-: *Fak*$^{CG}$ | (*Grabbe et al., 2004*) | N/A | |
| Genetic reagent (*Drosophila melanogaster*) | RSU1-: *ics*$^2$ | This paper | N/A | Null allele for *ics* (RSU1) |
| Genetic reagent (*Drosophila melanogaster*) | tensin-: *by*$^{33c}$ | (*Torgler et al., 2004*) | N/A | |
| Genetic reagent (*Drosophila melanogaster*) | vinculin-: ΔVinc | (*Klapholz et al., 2015*) | N/A | |
| Genetic reagent (*Drosophila melanogaster*) | filamin/cheerioΔC: cher$^{s24}$ | (*Huelsmann et al., 2016*) | N/A | |
| Genetic reagent (*Drosophila melanogaster*) | filamin/cheerio open: cher$^{open\ MSR}$ | (*Huelsmann et al., 2016*) | N/A | Viable line made by homologous recombination |
| Genetic reagent (*Drosophila melanogaster*) | filamin/cheerio closed: cher$^{closed\ MSR}$ | (*Huelsmann et al., 2016*) | N/A | Viable line made by homologous recombination |
| Genetic reagent (*Drosophila melanogaster*) | filamin/cheerio ΔMSR: cher$^{ΔMSR}$ | (*Huelsmann et al., 2016*) | N/A | Viable line made by homologous recombination |
| Genetic reagent (*Drosophila melanogaster*) | βPS*: mys$^{b27}$ | (*Kendall et al., 2011*) | N/A | |
| Genetic reagent (*Drosophila melanogaster*) | talin-: rhea$^{62}$ | (*Klapholz et al., 2015*) | N/A | |
| Genetic reagent (*Drosophila melanogaster*) | WASH-: Wash$^{185*}$ | (*Nagel et al., 2017*) | N/A | |
| Genetic reagent (*Drosophila melanogaster*) | βPS-GFP: mys-GFP | (*Klapholz et al., 2015*) | N/A | Viable line made by homologous recombination |
| Genetic reagent (*Drosophila melanogaster*) | GFP-talin | (*Klapholz et al., 2015*) | N/A | Viable line made by homologous recombination |

*Continued on next page*

*Continued*

| Reagent type (species) or Resource | Designation | Source or reference | Identifiers | Additional information |
|---|---|---|---|---|
| Genetic reagent (*Drosophila melanogaster*) | talin-GFP | (*Klapholz et al., 2015*) | N/A | Fully functional gene trap |
| Genetic reagent (*Drosophila melanogaster*) | Paxillin-GFP: | (*Bataillé et al., 2010*) | N/A | Genomic rescue construct |
| Genetic reagent (*Drosophila melanogaster*) | Git-GFP: | (*Bulgakova et al., 2017*) | N/A | Genomic rescue construct |
| Genetic reagent (*Drosophila melanogaster*) | ILK-GFP: | (*Zervas et al., 2001*) | N/A | Genomic rescue construct |
| Genetic reagent (*Drosophila melanogaster*) | PINCH-GFP: | (*Kadrmas et al., 2004*) | N/A | Genomic rescue construct |
| Genetic reagent (*Drosophila melanogaster*) | RSU1-GFP: | This paper | N/A | Genomic rescue construct |
| Genetic reagent (*Drosophila melanogaster*) | tensin-GFP: | (*Torgler et al., 2004*) | N/A | Genomic rescue construct |
| Genetic reagent (*Drosophila melanogaster*) | vinculin-GFP: | (*Klapholz et al., 2015*) | N/A | Genomic rescue construct |
| Genetic reagent (*Drosophila melanogaster*) | ZASP-GFP: | (*Jani and Schöck, 2007*) | N/A | Fully functional gene trap |
| Genetic reagent (*Drosophila melanogaster*) | α-actinin-GFP: | (*Buszczak et al., 2007*) | N/A | Fully functional gene trap |
| Genetic reagent (*Drosophila melanogaster*) | filamin/cheerio-GFP: cher-GFP | *Huelsmann et al., 2016*) | N/A | Viable line made by homologous recombination |
| Genetic reagent (*Drosophila melanogaster*) | UAS-WASH-GFP | (*Nagel et al., 2017*) | N/A | |
| Genetic reagent (*Drosophila melanogaster*) | zyxin-GFP: | This paper | N/A | Genomic rescue construct |
| Genetic reagent (*Drosophila melanogaster*) | ena-GFP: | Kindly provided by Yoshiko Inoue, Sven Huelsmann and Jenny Gallop | N/A | Genomic rescue construct |
| Genetic reagent (*Drosophila melanogaster*) | UAS-vinculin-RFP: | (*Maartens et al., 2016*) | N/A | |
| Genetic reagent (*Drosophila melanogaster*) | UAS-vinculinTail-RFP: | (*Maartens et al., 2016*) | N/A | |
| Genetic reagent (*Drosophila melanogaster*) | UAS-VBS2-GFP: | (*Maartens et al., 2016*) | N/A | |
| Genetic reagent (*Drosophila melanogaster*) | mef2-Gal4: P{Gal4-Mef2.R}3 | Bloomington Drosophila stock center | BDSC:50742 | |
| Genetic reagent (*Drosophila melanogaster*) | act88f-Gal4: P{Gal4-act88f} | (*Bryantsev et al., 2012*) | N/A | |

*Continued on next page*

*Continued*

| Reagent type (species) or Resource | Designation | Source or reference | Identifiers | Additional information |
|---|---|---|---|---|
| Genetic reagent (*Drosophila melanogaster*) | y1 P(nos-cas9, w+) M (3xP3-RFP.attP)ZH-2A w* | Bloomington Drosophila stock center | BDSC:54591 | |
| Genetic reagent (*Drosophila melanogaster*) | FAK-RNAi: | Bloomington Drosophila stock center | BDSC:44075 | |
| Genetic reagent (*Drosophila melanogaster*) | RSU1-RNAi: | Vienna Drosophila Resource centre | VDRC:42188 | |
| Genetic reagent (*Drosophila melanogaster*) | tensin-RNAi: | Vienna Drosophila Resource centre | VDRC:22823 | |
| Genetic reagent (*Drosophila melanogaster*) | filamin-RNAi: | Vienna Drosophila Resource centre | VDRC:107451 | |
| Genetic reagent (*Drosophila melanogaster*) | WASH-RNAi: | Vienna Drosophila Resource centre | VDRC:24642 | |
| Genetic reagent (*Drosophila melanogaster*) | UAS-ILK-GFP: | (*Zervas et al., 2011*) | N/A | |
| Genetic reagent (*Drosophila melanogaster*) | UAS-PINCHΔ1-GFP: | (*Zervas et al., 2011*) | N/A | |
| Genetic reagent (*Drosophila melanogaster*) | UAS-PINCHΔ4–5-GFP | This paper | N/A | |
| Genetic reagent (*Drosophila melanogaster*) | UAS-filamin90closed-RFPmito: | This paper | N/A | |
| Genetic reagent (*Drosophila melanogaster*) | UAS-filamin90open-RFPmito: | This paper | N/A | |
| Oligonucleotide | 5'-GTCGCTTCAAGAA CCCCATGTATG-3' | This paper | N/A | Guide RNA for mys-mcherry |
| Oligonucleotide | 5'-AAACCATACATGG GGTTCTTGAAG-3' | This paper | N/A | Guide RNA for mys-mcherry |
| Oligonucleotide | 5'-ACGCGTGGGAAT AGCAAACGCCACA-3' | This paper | N/A | Primer to amplify 5'UTR of zyxin |
| Oligonucleotide | 5'-TGTGTACTTGCG CATTCACA-3' | This paper | N/A | Primer to amplify 5'UTR of zyxin |
| Oligonucleotide | 5'-GACGTCAGAACA TTCGAGCTCATCGAT GAGTAAAGGA-3' | This paper | N/A | Primer to introduce Aat2 site to GFP |
| Oligonucleotide | 5' TGCGCAATAAATAAAAT GAGCACTCAATTTATTTGT ATAGTTCATCCATGC-3' | This paper | N/A | Primer to add Fsp1 site to GFP |
| Recombinant DNA reagent | pBluescript II KS | (addgene) | X52327.1 | |
| Recombinant DNA reagent | pBluescript II KS_mys-mcherry | This paper | N/A | Plasmid for generation of mys-mcherry flies |
| Recombinant DNA reagent | pCFD3 | (*Port et al., 2014*) | N/A | |
| Recombinant DNA reagent | BACR13D24 | (Berkeley Drosophila genome project) | AC010838 | |

*Continued on next page*

*Continued*

| Reagent type (species) or Resource | Designation | Source or reference | Identifiers | Additional information |
|---|---|---|---|---|
| Recombinant DNA reagent | TOPO TA | (Thermo Fisher Scientific) | 451641 | |
| Recombinant DNA reagent | pWhiteAttPRabbit zyxin-GFP | This paper | N/A | Plasmid for generation of zyxin-GFP flies |
| Recombinant DNA reagent | pWhiteAttPRabbit RSU1-GFP | This paper | N/A | Plasmid for generation of RSU1-GFP flies |
| Recombinant DNA reagent | pUASP-attB | Drosophila Genomics Resource center | 1358 | |
| Recombinant DNA reagent | MrRFPmito | (*Maartens et al., 2016*) | N/A | |
| Recombinant DNA reagent | pUASP-filamin90closed -RFPmito | This paper | N/A | Plasmid for generation of filamin90closed-RFPmito flies |
| Recombinant DNA reagent | pUASP-filamin90open -RFPmito | This paper | N/A | Plasmid for generation of filamin90open-RFPmito flies |
| Software | FIJI (ImageJ v1.5) | NIH | RRID:SCR_002285 | |
| Software | softWoRx 6.0 | Applied Precision | | |
| Software | Prism | GraphPad | RRID:SCR_002798 | |
| Software | Adobe Photoshop | Adobe | RRID:SCR_014199 | |
| Software | Microsoft Excel | Microsoft | RRID:SCR_016137 | |

## Fly husbandry

Flies were grown and maintained on food consisting of the following ingredients for 20 litres of food: 150 g agar, 1100 g glucose, 700 g wheat flour, 1000 g dry yeast, 500 ml nipagin 10%, 80 ml proprionic acid and 200 ml penicillin/streptomycin.

Animals of both sexes were used for this study with the exception of vinculin-; mef2 >vinculinTail RFP where only males were used. Expression of UAS::vinculin-RFP and UAS::vinculinTail-GFP (*Maartens et al., 2016*) were expressed in muscles with P{Gal4-Mef2.R}3 (Bloomington *Drosophila* stock centre, 54591). UAS::vinculinHead-RFP (*Maartens et al., 2016*) was expressed specifically in the IFMs using *Gal4-act88f* (*Bryantsev et al., 2012*), since expression with *Gal4-mef2* is lethal.

## Flight assays

Flight assays were performed as previously described (*Weitkunat and Schnorrer, 2014*). Briefly, flies were dropped into plastic tube (1 meter long and 15 cm wide) coated with a layer of oil, with water at the bottom. Flightless flies were counted as those that fall directly in the water and flying flies were counted as those that stick to the sides of the tube. 30 flies per condition were tested and percentage of flightless flies was calculated.

## Generation of a null RSU1 (*icarus*) allele, *ics*[2]

To ensure we had a null allele, a new *ics* mutant allele was generated by excision of the P-element insertion P{GT1}ics[BG02577], which is inserted 129 bp downstream of the translational start site. Two hundred single males of the genotype w; P{GT1}ics[BG02577]/CyO; mus309[N1]/mus309[D2] Sb P{Δ2,3}99B were crossed in individual vials to w; Sco/CyO; Dr/TM6 virgin females. From the progeny of each cross, three white-eyed excision males were individually crossed to w; Gla/CyO virgin females. Deletions were identified by PCR and characterized by sequencing. The allele *ics*[2] is a 1775 bp deletion which deletes 781 bp upstream and 994 bp downstream of the P-element insertion site, deleting the first 165 of 283 RSU1 residues.

## Generation of βPS-mcherry

βPS-mcherry was generated by CRISPR. A plasmid containing homology arms of *mys* (*Drosophila* gene encoding βPS) for the generation of βPS-GFP (*Klapholz et al., 2015*), was used as a template to generate a homology plasmid for the generation of βPS-mcherry. An 8502 bp fragment

containing 4465 pb of the C-terminal coding region of *mys*, GFP and 3894 bp of downstream non-coding sequence was cloned into pBluescript II KS (addgene) using Not1 and Xba1. GFP was excised by digestion with Hind3 and Sac1 and replaced with mcherry. A guide RNA targeting the C-terminus of βPS was cloned into the pCFD3 plasmid as previously described (**Port et al., 2014**). The sequence of the oligonucleotide used to generate the guide RNA were 5′-GTCGCTTCAAGAACCCCATGTA TG-3′ and 5′-AAACCATACATGGGGTTCTTGAAG-3′. The homology plasmid and the guide RNA plasmid were co-injected into *y*[1] *P{nos-cas9, w+} M{3xP3-RFP.attP}ZH-2A w** embryos. Injected adults were crossed to FM7 and the progeny were screened at larval stages by fluorescence microscopy for the presence of βPS-mcherry.

## Generation of zyxin-GFP

BACR13D24 (Berkeley *Drosophila* genome project) containing the genomic region of *Drosophila* zyxin was used as a template. A 9865 bp fragment containing the zyxin genomic region was excised from the BAC using Spe1 and Not1 and cloned into pBluescript II KS (addgene). A 1601bp region 5′ to the Spe1 site was PCR amplified from *Drosophila* genomic DNA and cloned into TOPO TA (Thermo Fisher Scientific). The forward primer added an Mlu1 site and had the sequence 5′-ACGCG TGGGAATAGCAAACGCCACA-3′, while the reverse primer was 5′-TGTGTACTTGCGCATTCACA-3′. pBluescript II KS containing the GFP sequence was used a template for GFP. A 738 bp product containing GFP was PCR amplified and cloned into TOPO TA. An Aat2 site was added with the forward primer 5′-GACGTCAGAACATTCGAGCTCATCGATGAGTAAAGGA-3′ and an Fsp1 site was added with the reverse primer 5′ TGCGCAATAAATAAAATGAGCACTCAATTTATTTGTATAGTTCATCCA TGC-3′. GFP was cloned into the 3′ of the zyxin genomic sequence at Fsp1 and Aat2 sites. The resulting 10,603 bp zyxin-GFP sequence was excised using Spe1 and Not1, and the 1601 bp 5′ region was excised from TOPO TA using Mlu1 and Spe1, and both were cloned into Mlu1 and Not1 sites of the vector pWhiteAttPRabbit (Brown, N and Klapholz, B unpublished). The vector contains attB sites and the white gene. The constructs were inserted into position 51 on chromosome II.

## Generation of RSU1-GFP

A wild type *ics* genomic rescue construct tagged with GFP, *RSU1-GFP*, was prepared by subcloning a 12,888 bp AatII to NotI genomic DNA fragment containing 6853 bp upstream of the translational start site of the *ics* locus into the targeted P-element transformation vector pWhiteAttPRabbit. To fuse GFP to the COOH terminus of RSU1, XhoI and HindIII sites were inserted between the last codon and the stop codon of the *ics* gene, and at the NH2 and COOH termini of GFP. A short linker amino acid sequence was introduced at the NH2 terminus of GFP so that the fusion protein junction corresponds to RA*YSSSS*MSK, with the linker underlined. The constructs were inserted into position 51 on chromosome II and recombined with *ics*[2].

## Generation of UAS-PINCHΔ4–5

Using the UAS-PINCH-GFP construct described in **Zervas et al., 2011** we deleted LIM domains 4 and 5 (residues 198–317 of isoform Stck-PA) and generated transgenic lines by P-element mediated tranformation.

## Generation of filamin-RFPmito constructs

Gibson cloning was used to clone a 2756 bp genomic fragment containing filamin90 and RFPmito into pUASP-attB (Drosophila Genomics Resource center). RFPmito was amplified from MrRFPmito plasmid (**Maartens et al., 2016**) using primers 5′-ccagatcgatgtcTCGAGCTCCAGCATGGTG-3′ and 5′- ttaacgttaacgttcgaggtcgactctagaACTAGTTGTTCTTGCGCAGTTG-3′ and filamin90 was amplified from pGE-attB-filaminopen-GFP or pGE-attB-filaminclosed-GFP (**Huelsmann et al., 2016**) using 5′- gtacccgcccggggatcagatccgcggccgcGGCCGCAAAATGACTACG-3′ and 5′- tgctggagctcgaGACA TCGATCTGGAATGG-3′. Fragments were incubated with pUASP-attB linearize with Xba1 and Not1.

## Immunohistochemistry

IFMs were dissected from pupae and adult flies as previously described (**Weitkunat and Schnorrer, 2014**). Briefly pupal thoraces were dissected in PBS and fixed in 4% formaldehyde in PBS + 0.3% Triton X-100 (PBST) for 15 min. Adult thoraces were dissected in PBS and fixed in 4% formaldehyde in

PBS for 1 hr. Thoraces were washed in PBST and dissected further as required. Thoraces were blocked in PBST +0.5% BSA and incubated in primary antibody overnight at 4°C. Primary antibodies used were rabbit anti-obscurin (*Burkart et al., 2007*) diluted 1:100, rat anti-filamin C-terminus (*Sokol and Cooley, 1999*) diluted 1:250 and rabbit anti-arp3 diluted 1:500 (*Stevenson et al., 2002*). Thoraces were incubated with anti-rabbit-Alex Fluor 488 (Abcam) if needed and phalloidin conjugated with rhodamine (Thermo Fisher Scientific), Dylight650 (Cell Signaling Technology) or Alex Fluor 488 (Thermo Fisher Scientific). IFMs were further dissected and mounted in Vectashield (VectorLabs).

## Microscopy

Confocal images were taken with an Olympus FV1000 confocal microscope using a 60×/1.35 NA objective and 3x zoom. A single stack of 5 z-sections spaced by 0.5 µm was imaged per individual. SIM images were taken with a DeltaVision OMX BLAZE V3 (Applied Precision) using an Olympus 10 × 0.4 NA air objective or 60 × 1.4 NA oil objective, 488 nm and 593 nm laser illumination and standard excitation and emission filter sets. Raw data was reconstructed using softWoRx 6.0 (Applied Precision) software. Images were processed using ImageJ (NIH) and Adobe Photoshop.

## Transmission electron microscopy

Half thoraces were dissected from 1 day old flies in PBS and fixed in glutaraldehyde/2% formaldehyde in 0.05 M sodium cacodylate buffer pH 7.4 for 4 hr at 4°C. Thoraces were osmicated in 1% $OsO_4$/1.5% potassium ferricyanide/0.05 M NaCAC pH 7.4 overnight at 4°C and then treated with 0.1% thiocarbohydrazide/DIW for 20 min. Thoraces were osmicated again in 2% $OsO_4$ for 1 hr and finally embedded in Quetol resin. 70 nm sections were imaged on a Tecnai G2 80-200kv transmission electron microscope at 6500x magnification.

## Quantification and statistical analysis

Phenotypes were analysed manually using ImageJ. The line tool was used to measure MTZ length, myofibril width, sarcomere length and the distance from the myofibril end to the first M-line. 10–30 myofibrils per individual were measured from 10 individuals per genotype. Average MTZ length of distance to first M-line was calculated per individual. Statistical significance was determined by Mann Whitney U test (p<0.05) using Prism software (GraphPad). Fluorescence intensity plots for SIM images were using the plot profile function in ImageJ (NIH). Profiles were processed and distance between peaks was calculated in Microsoft Excel.

Fluorescence intensity of GFP tagged proteins at the MTZ were quantified using a Matlab script adapted from *Bulgakova et al. (2017)*. Briefly, all objects in each frame were detected by a series of dilation, hole filling and eroding. The resulting objects were filtered by their area, eccentricity and orientation (more than 45° to long image axis) to exclude all objects that did not represent MTZs. MTZ minor axis length and mean intensities of the resulting objects were collected from original non-modified confocal frames and averaged. Mean intensity was then multiplied by the mean axis length to correct for differences in the MTZ length in different mutants. Statistical significance was determined by Mann Whitney U test (p<0.05) using Prism software (GraphPad).

## Contact for reagent and resource sharing

Further information and requests for resources and reagents should be directed to and will be fulfilled by the Lead Contact, Nick Brown (nb117@cam.ac.uk).

## Acknowledgements

We would like to thank Golnar Kolahgar and Benjamin Richier for critical reading of the paper; John Overton for performing microinjections; Nicola Lawrence for assistance with SIM imaging; Yoshiko Inoue, Sven Huelsmann, Jenny Gallop, Sven Bogdan, Frank Schnorrer, Richards Cripps and the Bloomington Stock Center for providing fly stocks; Lynn Cooley for antibodies; and Lyn Carter and Karin Muller from the Cambridge Advanced Imaging Centre for assistance with electron microscopy and sample preparation. This work was funded by a Wellcome Trust studentship (099739/Z/12/Z) to HG,

a Medical Research Council Studentship to AGMG, a BBSRC grant BB/L006669/1 to NB, a Acadamy of Finland grant (278668), and a Jenny and Antti Wihuri foundation research sabbatical grant to JY.

## Additional information

### Funding

| Funder | Grant reference number | Author |
|---|---|---|
| Wellcome | 099739/Z/12/Z | Hannah J Green |
| Suomen Akatemia | 278668 | Hannah J Green<br>Jari Ylänne |
| Medical Research Council | | Annabel G M Griffiths |
| Jenny ja Antti Wihurin Rahasto | Foundation research sabbatical grant | Jari Ylänne |
| Biotechnology and Biological Sciences Research Council | BB/L006669/1 | Nicholas H Brown |

The funders had no role in study design, data collection and interpretation, or the decision to submit the work for publication.

### Author contributions

Hannah J Green, Conceptualization, Formal analysis, Funding acquisition, Validation, Investigation, Visualization, Writing—original draft, Writing—review and editing; Annabel GM Griffiths, Resources, Investigation; Jari Ylänne, Conceptualization, Supervision, Funding acquisition, Writing—review and editing; Nicholas H Brown, Conceptualization, Supervision, Funding acquisition, Validation, Visualization, Writing—original draft, Writing—review and editing

### Author ORCIDs

Hannah J Green (iD) http://orcid.org/0000-0002-3039-3015
Jari Ylänne (iD) http://orcid.org/0000-0003-4627-021X
Nicholas H Brown (iD) http://orcid.org/0000-0002-8958-7017

### Decision letter and Author response

Decision letter https://doi.org/10.7554/eLife.35783.022
Author response https://doi.org/10.7554/eLife.35783.023

## Additional files

### Supplementary files

• Transparent reporting form
DOI: https://doi.org/10.7554/eLife.35783.019

### Data availability

All data generated or analysed during this study are included in the manuscript and supporting files

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
