## [Decision Letter]

Thank you for submitting your article "Novel functions for integrin-associated proteins revealed by myofibril attachment in *Drosophila*" for consideration by *eLife*. Your article has been reviewed by three peer reviewers, including Frank Schnorrer as the Reviewing Editor and Reviewer #1, and the evaluation has been overseen by Didier Stainier as the Senior Editor.

The reviewers have discussed the reviews with one another and the Reviewing Editor has drafted this decision to help you prepare a revised submission.

Summary:

Green et al., provide a comprehensive study of the distribution, and functional contribution of Integrin Associated Proteins (IAPs, i.e. FAK, RSU1, tensin and vinculin) at the myotendinous junction (MTJ) of the indirect flight muscles (IFMs). Previous experiments revealed surprisingly mild phenotypes in flies upon deletion of those proteins. The authors show that individual deletions of FAK, RSU1, tensin and vinculin lead to distinct morphological defects in IFMs, and genetic interaction studies suggest various degrees of interdependency, with positive and negative roles for integrin activity.

The experiments focus on a particular substructure of IFMs, called the modified terminal Z-disc (MTZ), which connects the myofibril to the neighboring tendon cell. The MTZ is affected to different extents by the loss of FAK, RSU1, tensin and vinculin. Unexpectedly, simultaneous deletion of FAK and RSU1 leads to the rescue of MTZ morphology, which is attributed to enhanced integrin activation (as a result of FAK depletion) since an active integrin mutant rescues the RSU1 phenotype as well. Activating integrins also partially rescued the phenotype of tensin-deficient flies, whereas FAK-overexpression leads to a tensin-mutant like MTZ morphology. The authors conclude from these experiments that FAK regulates integrin activity through tensin. The RSU1 phenotype can be (partially) rescued by reducing levels of its binding partner PINCH, whereas PINCH overexpression causes a RSU1-like MTZ phenotype; it is concluded that RSU1 inhibits PINCH activity. Next, the role of vinculin is analyzed and the authors find that depletion of this protein has two distinguishable consequences. First, loss of vinculin affects talin-dependent adhesion in the MTZ; second, it impairs actin organization in a newly defined structure called MARL (muscle actin regulatory layer). Additional genetic studies imply that vinculin regulates MARL organization through filamin, and MARL assembly requires the presence of the Arp2/3 regulator WASH.

Overall, the manuscript is interesting, the data are of high quality, and the amount of data is impressive. It adds important new structural information regarding the muscle-tendon junction structure (dividing it into 4 zones using super resolution and EM microscopy) and partially explains why so many IAPs are required to stably anchor myofibrils to tendons during the high force producing muscle contractions.

Some important points should be addressed during the revisions to clarify the introduction and to ideally further support the conclusions or tone-down/modify some of the molecular conclusions drawn in the paper. It will also be interesting to show the consequences of IAPs deletions for flight muscle function.

Essential revisions:

1) Specificity of the phenotypes. The authors convincingly show a defect of the terminal Z-disc structure in the IAP mutants. However, many IAPs are likely also expressed in tendon cells. Is the observed defect indeed solely due to a role of the IAPs in muscle? This is already convincingly shown for tensin using the muscle specific RNAi, but should be shown for the other IAPs as well. This will rule out an indirect effect of the function of the IAPs in tendons on the muscle attachment morphology. The authors have also tools ready to perform a muscle-specific rescue for the IAP mutants.

2) Functionality of GFP fusions. The authors gained important conclusions from their GFP-tagged IAP proteins. These are expressed from endogenous enhancers, thus over-expression artefacts are likely not an issue. However, the functionality of these GFP fusions should be tested wherever possible. Mutant rescue experiments with the IAP-GFP fusions used for the localisation studies appear straight forward. If not functional, it should be mentioned.

3) Molecular mechanisms. The paper several times states potentially interesting mechanisms of the IAPs that are not much supported by molecular data. Can the authors provide more convincing data to support how FAK inhibits integrin activity via tensin? How is integrin activity defined? Ligand binding affinity, or integrin dynamics, or mechanical interconnection with IAPs, etc.?

The authors state in the methods section that the βPS*: mys^B72^ mutant was used as previously described, but this line is not mentioned in the cited study (Kendall et al., 2011).

Alternatively, is there a way to support the hypothesis that RSU1 controls PINCH activity by molecular data? Pinch expression levels? IPP levels? IPP independent?

Or, can the intriguing hypothesis that vinculin promotes MARL formation through inducing a conformational change in filamin be supported by molecular data? Can the authors show a direct association between vinculin and filamin (e.g. through immunoprecipitation, FRET studies, or biochemical assays)?

4) Introduction/context. Some points in the intro should be made clearer. The development of the adult muscles in covered only very briefly in Dobi et al., 2015. A more appropriate recent review would be Gunage et al., 2017.

It is important for the remaining parts of the paper to clearly state that IFMs are "fibrillar" muscles, with individual myofibrils attaching to tendons, in contrast to the normal "tubular" cross-striated adult muscles. See Josephson et al., or Schonbauer et al.,. This might explain the exclusive phenotype in the IFMs for the IAP mutants.

Indirect flight muscles are built by multiple myofibers (not myofibrils as currently stated in the intro) and thus more similar to vertebrate muscles. Each myofiber contains many hundred myofibrils.

The sarcomere not only consists of "interdigitated" actin and myosin filaments, which can be present in many cell types, not only in muscles. Please describe the sarcomere structure more accurately. "Interdigitations" are also used throughout the text to describe defects in sarcomere length ("longer interdigitations", longer sarcomeres would be more appropriate.)

5) Force sensing structure. The description of the MARL domain is very interesting and novel. The authors suggest that this domain is induced by mechanotransduction. Can the authors reinforce this hypothesis by manipulating the force transmitted by the MTJ? One option might be to reduce myosin force by expressing a motor deficient version of myosin heavy chain under act88F control (Weitkunat et al., 2014).

Alternatively, they could try to reduce integrin function selectively in the tendon cells, which might make attachments less rigid.

6) Flight muscle function. Please show the results from the flight tests at day 1 and day 14 of the IAP mutants instead of quoting "unpublished results". It is surprising that these mutants can fly normally despite the defective attachments, in particular at day 14 of age.

[Editors' note: further revisions were requested prior to acceptance, as described below.]

Thank you for resubmitting your work entitled "Novel functions for integrin-associated proteins revealed by analysis of myofibril attachment in *Drosophila*" for further consideration at *eLife*. Your revised article has been favorably evaluated by Didier Stainier (Senior Editor), and Frank Schnorrer (Reviewing Editor) after consultation with two additional reviewers.

The manuscript has been improved but there are some remaining issues that need to be addressed before acceptance, as outlined below:

1) The authors state that vinculin mutants can fly (see figure legend, old point 6). However, the day 14 vinculin mutant shows severe myofibril defects (Figure 4A) and only about 20% of the flies appear to fly (Figure 4B). This should be a very robust difference that should be mentioned (p-value?).

2) Interdigitations (old point 4). Sarcomeres are now explained correctly and not described as "interdigitated" actin and myosin filaments. However, "interdigitations" are used to describe the folded electron dense structure/membrane at the myofibril tips (zone 1 of the MTZ). This is used first in subsection “Indirect flight muscles (IFMs) reveal phenotypes for viable IAP mutants” without a clear description of what these interdigitations mean. Please explain it better, including its appearance in the model (Figure 8). What do "longer interdigidations" in the model mean? Longer than what?

3) 'Web-like' structure of actin in SIM imaging (old point S4). If this is an artifact of the technique, it should be mentioned, at least in the figure legend.

4) In many of the sarcomere stains with phalloidin (not only the former rescued vinculin mutant image initially mentioned) for example the recused FAK or tensin mutant (Figure 1—figure supplement 1) or many of the GFP-fusion genotype images in Figure 2 and Figure 2—figure supplement 2 the characteristic high concentration of phalloidin at the Z-disc is not visible. This likely due to an artifact when cutting the myofibrils during histology (these are under high tension). As sarcomere length was not determined in these samples the issue appears not too large. However, we recommend fixing these artifacts wherever possible. The use of relaxing solution during the preparation should also help. A clean, parallel cut is important to preserve myofibrils. Cut myofibrils should wherever possible not be imaged.

5) Arrows in Figure 1A are yellow not white (legend). Same for Figure 7A. Figure 6: Obscurin/Unc-89 not Unc-86 (legend).

6) Weitkunat et al., 2014 do not show any spontaneous muscle contractions during pupal stages of the developing flight muscles (currently stated in the Introduction). Spletter et al., 2018 found these contractions, so this could be cited.

7) Subsection “Genetic interactions reveal inhibitory action and compensation between IAPs”: "RSU1 and integrin keep PINCH levels at the correct level; in its absence the […]"; in which absence? PINCH?

---

## [Author Response]

Essential revisions:1) Specificity of the phenotypes. The authors convincingly show a defect of the terminal Z-disc structure in the IAP mutants. However, many IAPs are likely also expressed in tendon cells. Is the observed defect indeed solely due to a role of the IAPs in muscle? This is already convincingly shown for tensin using the muscle specific RNAi, but should be shown for the other IAPs as well. This will rule out an indirect effect of the function of the IAPs in tendons on the muscle attachment morphology. The authors have also tools ready to perform a muscle-specific rescue for the IAP mutants.

We have performed muscle specific RNAi of all the IAPs and all of these phenocopy the mutant phenotype. This data can be found in Figure 1—figure supplement 1 and Figure 7—figure supplement 1. We do not show RNAi of vinculin as we had already shown muscle specific rescue in Figure 4—figure supplement 4.

2) Functionality of GFP fusions. The authors gained important conclusions from their GFP-tagged IAP proteins. These are expressed from endogenous enhancers, thus over-expression artefacts are likely not an issue. However, the functionality of these GFP fusions should be tested wherever possible. Mutant rescue experiments with the IAP-GFP fusions used for the localisation studies appear straight forward. If not functional, it should be mentioned.

The GFP-tagged proteins are expressed from 'genomic rescue constructs' or insertions into the endogenous locus (by gene trapping or homologous recombination), and all have previously been shown to rescue mutant alleles fully, or if at the endogenous locus, not cause a phenotype when homozygous viable. We have added a line stating this in the Results section.

3) Molecular mechanisms. The paper several times states potentially interesting mechanisms of the IAPs that are not much supported by molecular data. Can the authors provide more convincing data to support how FAK inhibits integrin activity via tensin?

We have not been able to advance our understanding of the details of this mechanism within the revision time.

How is integrin activity defined? Ligand binding affinity, or integrin dynamics, or mechanical interconnection with IAPs, etc.?The authors state in the methods section that the βPS*: mys^B72^ mutant was used as previously described, but this line is not mentioned in the cited study (Kendall et al., 2011).

It is b27, not b72. The mys^b27^ allele is discussed at length in Kendall et al., 2011 (e.g. Table1, hybrid domain mutants, and the section on Hybrid Domain Activation Mutants). In this paper the authors show that this allele has "extremely elevated affinity" for ligand, testing by putting this mutant form of integrin into cells in culture and measuring binding to the ligand mimetic antibody TWOW. The similarity in loss of FAK and the mys^b27^ allele rescue of RSU1 and vinculin is what supports the idea that FAK is negatively regulating integrin activity. We lack antibodies that report on *Drosophila* integrin activity within tissues (TWOW does not work for this), so unfortunately, we do not have a way of directly assessing the amount of active integrin in *Drosophila* tissues.

Alternatively, is there a way to support the hypothesis that RSU1 controls PINCH activity by molecular data? Pinch expression levels? IPP levels? IPP independent?

We have added additional data to address this mechanism, but have not got as far as fully solving it. We have quantified IAP levels in Figure 2 and included this in the revised Figure. This showed that in the RSU1 mutant Paxillin is elevated 2.5 fold (as mentioned in the text previously), ILK levels are reduced 2 fold, but PINCH levels did not change. We followed up to test whether elevating Paxillin would mimic elevation of PINCH, but it did not and instead produced a new phenotype that we felt was outside the scope of the paper. We then tested two truncation mutants of PINCH, for their ability to mimic the RSU1 mutant phenotype when overexpressed, as full-length PINCH does. Deletion of LIM4,5 reduced its function, but we still got a phenotype. This rules out a model where RSU1 acts by competing with another protein that binds to LIM5. We also found that overexpression of PINCH ∆LIM1 did not cause a phenotype, suggesting that PINCH still needs to interact with ILK to produce its gain of function phenotype. We have included this data in the Results section, and in a revised Figure 3—figure supplement 1 and modified the model in Figure 3.

Or, can the intriguing hypothesis that vinculin promotes MARL formation through inducing a conformational change in filamin be supported by molecular data? Can the authors show a direct association between vinculin and filamin (e.g. through immunoprecipitation, FRET studies, or biochemical assays)?

Yes, we have been able to show an association between the vinculin tail and the filamin C-terminus that is independent of whether filamin is open or closed (described in subsection “Filamin and WASH contribute to MARL formation” and included in Figure 8). For this we have used an assay where we target the filamin C-terminal 9 Filamin repeats (the isoform filamin90) to the mitochondria and show that it recruits vinculin tail, as well as actin and WASH. We prefer this to the IP as it shows they can interact within the intact cell, but like IP it does not distinguish direct from indirect. We tested whether purified filamin MSR region binds to the vinculin tail in vitro and it does not but have were not able to make and test other parts of filamin90. FRET would be a good approach, but not possible within the timeframe of revisions as we would need to map the interactions precisely enough to bring the FRET pairs in proximity and generate appropriate transgenic animals. The ability of filamin90 to recruit WASH made us reconsider whether they are in the same or different pathways and we realized that the majority of the data is consistent with WASH being downstream of mechanically opened filamin.

We have also been able to strengthen our analysis with TEM pictures for vinculin, filamin and WASH, which give a clearer view of the impact of the loss of these proteins on the structure of the MARL.

We thought it would be helpful to assemble all the data into a model that while speculative is consistent with all our data, which is described in subsection “Filamin and WASH contribute to MARL formation” and shown in Figure 8.

4) Introduction/context. Some points in the intro should be made clearer. The development of the adult muscles in covered only very briefly in Dobi et al., 2015. A more appropriate recent review would be Gunage et al., 2017.It is important for the remaining parts of the paper to clearly state that IFMs are "fibrillar" muscles, with individual myofibrils attaching to tendons, in contrast to the normal "tubular" cross-striated adult muscles. See Josephson et al., or Schonbauer et al.,. This might explain the exclusive phenotype in the IFMs for the IAP mutants.Indirect flight muscles are built by multiple myofibers (not myofibrils as currently stated in the intro) and thus more similar to vertebrate muscles. Each myofiber contains many hundred myofibrils.The sarcomere not only consists of "interdigitated" actin and myosin filaments, which can be present in many cell types, not only in muscles. Please describe the sarcomere structure more accurately. "Interdigitations" are also used throughout the text to describe defects in sarcomere length ("longer interdigitations", longer sarcomeres would be more appropriate.

Thank you for these suggestions, we have modified the text accordingly.

5) Force sensing structure. The description of the MARL domain is very interesting and novel. The authors suggest that this domain is induced by mechanotransduction. Can the authors reinforce this hypothesis by manipulating the force transmitted by the MTJ? One option might be to reduce myosin force by expressing a motor deficient version of myosin heavy chain under Act88F control (Weitkunat et al., 2014).

We did this experiment but found that the sarcomeres are so disrupted that it was difficult to make any firm conclusions (see Author response image 1). The MTZ does appear smaller, but whether this is due to loss of force alone would be difficult to conclude. We did not think this result substantially supported the idea that mechanical force is needed to build the MARL, so we have not added it to the revised version but let us know if you think it would be important to include it.

Alternatively, they could try to reduce integrin function selectively in the tendon cells, which might make attachments less rigid.

We appreciate the suggestion but think it would require extensive work to demonstrate that it was an alteration of forces that was having any effect observed.

6) Flight muscle function. Please show the results from the flight tests at day 1 and day 14 of the IAP mutants instead of quoting "unpublished results". It is surprising that these mutants can fly normally despite the defective attachments, in particular at day 14 of age.

Surprising but true, now quantified in Figure 1—figure supplement 1, Figure 2 and Figure 7—figure supplement 1. Only loss of filamin caused a substantial reduction in the number of flies that will fly in the test, and this is likely to be due to the defects in sarcomere structure as recently reported by Frieder Schöck and colleagues.

[Editors' note: further revisions were requested prior to acceptance, as described below.]1) The authors state that vinculin mutants can fly (see figure legend, old point 6). However, the day 14 vinculin mutant shows severe myofibril defects (Figure 4 A) and only about 20% of the flies appear to fly (Figure 4B). This should be a very robust difference that should be mentioned (p-value?).

We have revised subsection “Vinculin has two distinct functions in the MTZ” as follows:

“One day old flies lacking vinculin were able to fly normally but after 14 days, only 20% were able to fly (Figure 4B). However, muscle specific knockdown of vinculin did not reduce flying ability, yet caused comparable IFM actin defects, indicating that loss of flight in flies lacking vinculin is not due to the IFM defects.”

Unfortunately, we were not able to perform sufficient numbers of independent repeats to merit a p-value.

2). Interdigitations (old point 4). Sarcomeres are now explained correctly and not described as "interdigitated" actin and myosin filaments. However, "interdigitations" are used to describe the folded electron dense structure/membrane at the myofibril tips (zone 1 of the MTZ). This is used first in subsection “Indirect flight muscles (IFMs) reveal phenotypes for viable IAP mutants” without a clear description of what these interdigitations mean. Please explain it better, including its appearance in the model (Figure 8). What do "longer interdigidations" in the model mean? Longer than what?

We have added a sentence defining the interdigitations in the Introduction. We have also added ‘interdigitations’ to the model in Figure 5 and Figure 9. Finally, we have added “longer than wildtype” to the legend for Figure 9.

3) 'Web-like' structure of actin in SIM imaging (old point S4). If this is an artifact of the technique, it should be mentioned, at least in the figure legend.

We have added a sentence to the Figure 5 legend.

4) In many of the sarcomere stains with phalloidin (not only the former rescued vinculin mutant image initially mentioned) for example the recused FAK or tensin mutant (Figure 1—figure supplement 1) or many of the GFP-fusion genotype images in Figure 2 and Figure 2—figure supplement 2 the characteristic high concentration of phalloidin at the Z-disc is not visible. This likely due to an artifact when cutting the myofibrils during histology (these are under high tension). As sarcomere length was not determined in these samples the issue appears not too large. However, we recommend fixing these artifacts wherever possible. The use of relaxing solution during the preparation should also help. A clean, parallel cut is important to preserve myofibrils. Cut myofibrils should wherever possible not be imaged.

As pointed out, the morphology of the sarcomeres is not the focus of the paper, and we have not noted any effects on the attachment structure, nor IAP distribution, of the contractile state of the sarcomeres. For example, we did not notice any changes when dissecting in relaxing solution. Nonetheless, to provide images that are as similar as possible we have replaced those images with a dark Z-line with available alternative images and did new stainings to replace the rescued FAK mutant (Figure 1—figure supplement1, Figure 2 and Figure 2—figure supplement 1).

5) Arrows in Figure 1A are yellow not white (legend). Same for Figure 7A. Figure 6: Obscurin/Unc-89 not Unc-86 (legend).

Thanks. These have now been fixed in the figure legends.

6) Weitkunat et al., 2014 do not show any spontaneous muscle contractions during pupal stages of the developing flight muscles (currently stated in the Introduction). Spletter et al., 2018 found these contractions, so this could be cited.

Sorry, our mistake, we were meaning to say the increased tension as measured in Weitkunat et al., 2014.

7) Subsection “Genetic interactions reveal inhibitory action and compensation between IAPs”: "RSU1 and integrin keep PINCH levels at the correct level; in its absence the […]"; in which absence? PINCH?

We have removed “its” and replaced with “in the absence of RSU1”.